# Effect of Geometry on Graph Neural Networks

## Abstract

Hyperbolic Graph Neural Networks (GNNs) have emerged as a promising approach for modeling graph-structured data with less embedding distortion than Euclidean GNNs. In this paper, we explore the effect of geometry on the performance of three types of GNNs for node classification and link prediction. To do so, we build on the hyperbolic framework outlined in Chen et al. (2022) and propose a family of GNNs with alternating geometry, integrating both hyperbolic and Euclidean components that can be trained jointly. We compare our alternating geometry models' performance and stability against their Euclidean and hyperbolic counterparts across various datasets. Finally, we examine the impact of the choice of geometry and graph properties on hyperparameter selection. The alternating geometry models achieved the best performance in node classification, while the hyperbolic models outperformed alternating and Euclidean models in link prediction. Additionally, for node classification, architecture choice had a greater impact on performance than geometry, whereas for link prediction, geometry had a more significant effect than architecture.

## 1 Introduction

With the advent of Graph Neural Networks (GNNs), geometric deep learning has gained significant prominence (Bronstein et al., 2021). GNNs learn functions on data that contain a graph $G = (V, E)$ and feature vectors $x_i$ at each node. The idea is that, in addition to the feature vectors, the graph structure $G$ captures geometric structure present in the data. One example of such a geometric structure is curvature. Recently, for nearest neighbor graphs constructed from data on a manifold, Trillos & Weber (2023) showed that the discrete Ricci curvature on the graph (Ollivier, 2009) is consistent with the continuous Ricci curvature on the manifold. Thus, the graph structure provides valuable geometric information.

GNNs process this geometric information using message passing. Concretely, this consists of two steps. First, the feature vectors at each node are mapped by a neural network $F$ to new feature vectors. Conventionally, an identical neural network is applied to vectors at all nodes. We refer to this step as *feature extraction*. After the feature extraction step, GNNs aggregate information from neighboring nodes. The simplest form of aggregation is to compute the weighted average of the neighboring feature vectors. This step is referred to as *aggregation*.

Recent work has shown that parameterizing the feature extraction neural network using hyperbolic geometry improves performance (Chen et al., 2022; Chami et al., 2019; Zhang et al., 2021b;a). This approach involves factoring the neural network through the hyperbolic manifold rather than learning a direct map from Euclidean space to Euclidean space. These methods first embed the feature vectors into hyperbolic space, perform feature extraction and aggregation in hyperbolic

space, and finally map back to Euclidean space to obtain the class label. *In this paper, we investigate how different geometric parameterizations of neural networks affect GNN performance.* Our study examines various geometric approaches, compares their effectiveness, and analyzes the implications for GNN architecture design.

## 1.1 Different Geometries

The geometry referenced above is just one of the types of geometries associated with neural networks. Most works in geometric machine learning literature focus on one of the following different types of geometries.

**Definition 1** (Data Geometry). *Data Geometry refers to the geometry of the manifold that the data is supported on.*

**Definition 2** (Function Space Geometry). *Function Space Geometry is the geometry of the space of functions that can be modeled by the neural network.*

**Definition 3** (Parameter Geometry). *Parameter Geometry is the geometry of the space of parameters.*

It is important to note that despite the relationship between parameterized functions and their parameter spaces, the geometry of the parameter space may differ from that of the function space.

We consider two additional geometries. The first is feature geometry, which, while not always apparent, is commonly studied in prior works such as Shimizu et al. (2020); Ganea et al. (2018); Lensink et al. (2022); Chen et al. (2022); Chami et al. (2019); Zhang et al. (2021a); Liu et al. (2019); Zhang et al. (2021b). In these works, when they define hyperbolic (graph) neural networks, the term "hyperbolic" refers to the geometry of the feature space.

**Definition 4** (Feature Geometry). *Feature Geometry is the geometry of the manifold on which the pre/post activations of each layer reside.*

The final geometry type is the metric induced by the loss function.

**Definition 5** (Loss Geometry). *Given two parameterized functions $f(x; \theta_1), f(x; \theta_2)$ (or just two functions $f_1, f_2$ in function space), the metric induced by a loss function $\ell$, and data distribution $\rho$ is*

$$d(\theta_1, \theta_2) := \mathbb{E}_{x \sim \rho(x)} \left[ \ell(f(x; \theta_1), f(x; \theta_2)) \right]$$

Each of these five different geometries contributes significantly to the model's implicit bias, influencing how the neural network learns and generalizes from data. The interplay between these geometries is complex and often subtle, but fundamentally shapes the behavior and performance of neural networks. Data geometry informs us about the structure of the input space, while function space geometry determines the expressiveness of our model. Parameter geometry and loss geometry affect optimization dynamics, and feature geometry influences the internal representations learned by the network.

Understanding these geometries and their interactions is crucial for designing more effective and efficient neural network architectures, especially in the context of Graph Neural Networks. By carefully considering and manipulating these geometries, we can potentially enhance model performance, improve generalization, and tackle complex problems in geometric deep learning more effectively.

## 1.2 Main Question and Contribution

In this paper, we make several contributions to the understanding and application of these geometries in the context of Graph Neural Networks. Our primary focus is investigating whether feature geometry results in global improvements, independent of other factors such as architecture type, downstream task, or dataset. It is important to note that these additional factors are also associated with their respective geometries. Specifically, the model architecture influences both the function space geometry and the parameter space geometry. The downstream task is inherently tied to the loss geometry, whereas the dataset is fundamentally connected to the data geometry.

To conduct this investigation, we introduce an intermediary between Euclidean and hyperbolic networks, which we term "alternating geometry neural networks." We then evaluate the effect of feature geometry through a comprehensive study. This involves considering three distinct model architectures for each geometry and assessing these nine models (three geometries multiplied by three architectures) across two tasks and nine different datasets. This extensive evaluation allows us to isolate the impact of feature geometry and understand its interaction with other geometric aspects of the learning process.

**Contributions**  Our main contributions are

1. We demonstrate that the effect of feature geometry is highly task-dependent. Specifically, we observe distinct trends for node classification and link prediction tasks.

2. We show that alternating geometry models exhibit superior performance for node classification tasks, while hyperbolic models excel in link prediction tasks.

3. We reveal that for node classification, the model's architecture has a more significant impact on performance than the geometry. In contrast, for link prediction tasks, feature geometry plays a more crucial role in determining performance.

4. We provide a comprehensive dataset comprising of training, validation, and test statistics for each epoch of over 50,000 trained neural networks. This extensive dataset serves as a valuable resource for further analysis of geometry's impact on neural network performance.

The remainder of this paper is structured as follows. Section 2.1 provides background on hyperbolic models, establishing the theoretical foundation for our work. Section 3 introduces our alternating geometry models, detailing their construction and properties. Section 4.1 describes the datasets and models employed in our experiments. Section 5 presents the results and analysis of our node classification experiments. Section 6 focuses on the link prediction experiments, discussing our findings and their implications.

## 1.3 Prior Work

Various types of data can be represented in hyperbolic space with smaller distortion than Euclidean space. This advantage extends to diverse domains such as text (Dhingra et al., 2018), social and other networks (Krioukov et al., 2009; Shavitt & Tankel, 2008; Boguná et al., 2010), cell development trees (Klimovskaia et al., 2020), and even computer vision (Khrulkov et al., 2020; Desai et al., 2023). Generally, data with rich hierarchical semantic structures benefit from hyperbolic representations. For a comprehensive overview of applications, readers are directed to surveys by Yang et al. (2022); Peng et al. (2021); Mettes et al. (2023).

These successes have spurred significant research in developing methods for embedding data in hyperbolic space (Sonthalia & Gilbert, 2020; Nickel & Kiela, 2017; 2018; Sala et al., 2018; Lin et al., 2023). Concurrently, there has been substantial progress in developing hyperbolic neural networks (Shimizu et al., 2020; Ganea et al., 2018; Lensink et al., 2022; Chen et al., 2022) and hyperbolic graph neural networks (Chami et al., 2019; Zhang et al., 2021a; Liu et al., 2019; Zhang et al., 2021b).

While hyperbolic and Euclidean geometries have shown promise in various domains, recent research has explored the potential of mixed curvature models. This exploration is particularly relevant given that graph datasets often contain regions with both flat and negative curvature, as demonstrated by Tortorella & Micheli (2023). Mixed curvature models combine different geometric spaces to better capture complex data structures. Gu et al. (2019) introduced a framework for learning embeddings in a product manifold of constant curvature spaces, demonstrating improved performance on several benchmark datasets. In the context of graph neural networks, Wang et al. (2021) developed a mixed-curvature GNN that adaptively learns the appropriate curvature for different subgraphs. These works highlight the potential of mixed curvature approaches to capture intricate geometric properties of data that may not be adequately represented by a single geometric space. Building on this foundation, recent research has explored spaces with varying curvature or combinations of geometries as alternative frameworks for neural networks (Sonthalia et al., 2022; Zhao et al., 2023; Xu et al., 2022; López et al., 2021; Lopez et al., 2021). These studies further underscore the importance of considering diverse geometric representations in machine learning models.

## 2 Neural Networks in Different Geometries

Let $G = (V, E)$ be a graph, where $n = |V|$ is the number of nodes and $m = |E|$ is the number of edges. Let $X \in \mathbb{R}^{d \times n}$ be a matrix of *feature vectors*, where the $i^{th}$ column of $X$ corresponds to a $d$-dimensional feature vector for the $i^{th}$ node. The pair $(X, G)$ can be thought of as an annotated graph. Graph neural networks (GNNs) are techniques for parameterizing maps from the space of annotated graphs to regression targets or class labels. There are three main tasks:

1. **Node Classification:** Given an annotated graph $(X, G)$, we learn a map that assigns a class label to each node. Typically, labels are known for a subset of nodes, and the network predicts labels for the remaining nodes.

2. **Link Prediction:** Given an annotated graph $(X, G)$, we assume the existence of a supergraph $H$ such that $G$ is a subgraph of $H$. The task is to learn a map that predicts, for any pair of nodes, whether they are connected in $H$.

3. **Graph Classification/Regression:** We learn a map from a family of annotated graphs to either a vector (regression) or a class label (classification) for each graph.

All these maps are parameterized similarly. Analogous to standard neural networks, we stack multiple GNN layers to construct the desired map. Each GNN layer consists of two primary components: the *feature mapping* step and the *aggregation* (or *message passing*) step.

## 2.1 Hyperbolic Graph Neural Networks

We briefly introduce the Hyperboloid or Lorentzian model of hyperbolic manifolds. First, we define the Lorentzian inner product. Given $x, y \in \mathbb{R}^{k+1}$:

$$\langle x, y \rangle_{\mathcal{L}} = -x_0 y_0 + \sum_{i=1}^{k} x_i y_i$$

The Lorentzian model $\mathbb{L}^k$ of the hyperbolic manifold with constant negative curvature $-K$ is defined as

$$\mathbb{L}^k = \left\{ x \in \mathbb{R}^{k+1} : x_0 > 0, \langle x, x \rangle_{\mathcal{L}} = -\frac{1}{K} \right\}. \tag{1}$$

Distances in this model are given by

$$d(x, y) = \operatorname{arccosh} \left( -\sum_{i=1}^{n} x_i y_i + x_{n+1} y_{n+1} \right).$$

For this model, we have explicit formulas for the Exponential and Logarithmic maps. Specifically, the Exponential map is given by

$$\operatorname{Exp}_x(v) = \cosh(\|v\|_{\mathcal{L}})x + \sinh(\|v\|_{\mathcal{L}}) \frac{v}{\|v\|_{\mathcal{L}}},$$

where $\|x\|_{\mathcal{L}}^2 = -x_0^2 + \sum_{i=1}^{k} x_i^2$ and the Logarithmic map is given by

$$\operatorname{Log}_x(y) = \frac{\operatorname{arccosh}\left(\langle x, y \rangle_{\mathcal{L}}\right)}{\sqrt{\langle x, y \rangle_{\mathcal{L}}^2 - 1}} \left( y - \langle x, y \rangle_{\mathcal{L}} x \right).$$

### 2.1.1 Using Tangent Space.

One standard method of defining hyperbolic (graph) neural network layers uses the tangent space of the manifold. Let $F$ be any (graph) neural network layer on Euclidean space. Then, the corresponding hyperbolic version is given as follows

$$HF(x) = \operatorname{Exp}_{x_0}\left(F\left(\operatorname{Log}_{x_0}(x)\right)\right)$$

where Exp, Log are the Exponential and Logarithmic maps for the manifold, and $x_0$ is a base point. In some cases, only the feature mapping is done in the tangent space, and the aggregation is done in hyperbolic using the Frechet mean. This approach is often computationally expensive. Works such as Chami et al. (2019); Zhang et al. (2021a); Liu et al. (2019); Zhang et al. (2021b) broadly fall into this category.

### 2.1.2 Fully Hyperbolic Models

Chen et al. (2022) introduced a fully hyperbolic framework that adopts the Lorentz model as its feature space. Since the manifold is embedded into Euclidean space, given a vector $x \in \mathbb{L}^k \subset \mathbb{R}^{k+1}$, we can apply $W \in \mathbb{R}^{k+1 \times k+1}$ to $x$ to get a vector $Wx$ in $\mathbb{R}^{k+1}$. However, $Wx$ need not lie on the Lorentzian manifold. The linear transformations $W$ that map the Lorentz model to itself are called the Lorentz transformations. Moretti (2002) showed that these transformations can be decomposed into a Lorentz rotation and a Lorentz boost. Lorentz rotations characterize rotation in spatial coordinates, whereas the Lorentz boosts describe relative motion with constant velocity.

**Definition 6** (Lorentz Rotation)**.** *The Lorentz rotation matrices are given by*

$$\mathbf{R} = \begin{bmatrix} 1 & \mathbf{0}^\top \\ \mathbf{0} & \tilde{\mathbf{R}} \end{bmatrix}, \ where \ \tilde{\mathbf{R}}^\top \tilde{\mathbf{R}} = \mathbf{I}, \ det(\tilde{\mathbf{R}}) = 1.$$

**Definition 7** (Lorentz Boost)**.** *Given a velocity* $\mathbf{v} \in \mathbb{R}^n, \|\mathbf{v}\| < 1$ *and* $\gamma = \frac{1}{\sqrt{1-\|\mathbf{v}\|^2}}$*, the Lorentz boost matrices are given by*

$$\mathbf{B} = \begin{bmatrix} \gamma & -\gamma \mathbf{v}^\top \\ -\gamma \mathbf{v} & \mathbf{I} + \frac{\gamma^2}{1+\gamma} \mathbf{v}\mathbf{v}^\top \end{bmatrix}$$

Chen et al. (2022) prove that a hyperbolic linear layer that relies on logarithmic maps and exponential maps can only model a special form of "pseudo-rotation" but cannot model the Lorentz boost. Hence, they propose the following hyperbolic linear layer that covers both Lorentz rotation and boost. Given a matrix $M = \begin{bmatrix} v \\ W \end{bmatrix}$ and an input $x$, the output to the hyperbolic linear layer is given by

$$z := \begin{bmatrix} \frac{\sqrt{\|Wx\|^2+1}}{v^T x} v \\ W \end{bmatrix} x.$$

They show that if $x$ resides in the Lorentz manifold, then $z$ must also reside in the Lorentz manifold. These models are trained using Riemannian Adam (Kochurov et al., 2020).

## 3   Alternating Geometry Networks

To better understand the impact of the choice of geometry, we define a neural network that can be interpreted as living between Euclidean and hyperbolic neural networks. We do this by simplifying the linear layer from Chen et al. (2022). Specifically, we define a linear layer $HL$ that maps $\mathbb{R}^d \to \mathbb{L}^k$ as follows: Given a matrix $W \in \mathbb{R}^{k \times d}$ and a vector $x \in \mathbb{R}^d$, define $\hat{x} := Wx \in \mathbb{R}^k$. Then the output $z := HL(x; W) \in \mathbb{L}^k \subset \mathbb{R}^{k+1}$, is given by

$$z = \begin{bmatrix} z_0 \\ \hat{x} \end{bmatrix},$$

where

$$z_0 := \sqrt{1 + \|\hat{x}\|^2}. \tag{2}$$

This layer differs slightly from the one in Chen et al. (2022) as it does not involve the vector $v$. Importantly, this layer maps not only the Lorentz manifold to itself but *all of* $\mathbb{R}^d$ to the Lorentz manifold, as shown in the following proposition.

**Proposition 8.** *For any vector* $x \in \mathbb{R}^d$*, and any matrix* $W \in \mathbb{R}^{k \times d}$ *the output* $z = HL(x; W)$ *lives in the Hyperboloid manifold* $\mathbb{L}^k$*.*

*Proof.* By construction, $z_0^2 - \sum_{i=1}^k z_i^2 = 1$, satisfying (1). Thus, $z \in \mathbb{L}^k$. □

Unlike the model proposed by Chen et al. (2022), this method eliminates the need for specialized aggregation steps in GNNs or custom non-linear activations that preserve manifold structure. Specifically, this approach additionally allows for a more straightforward implementation. After applying $z = HL(x^{(0)}; W)$, we can simply use any standard nonlinear activations $f$ element-wise

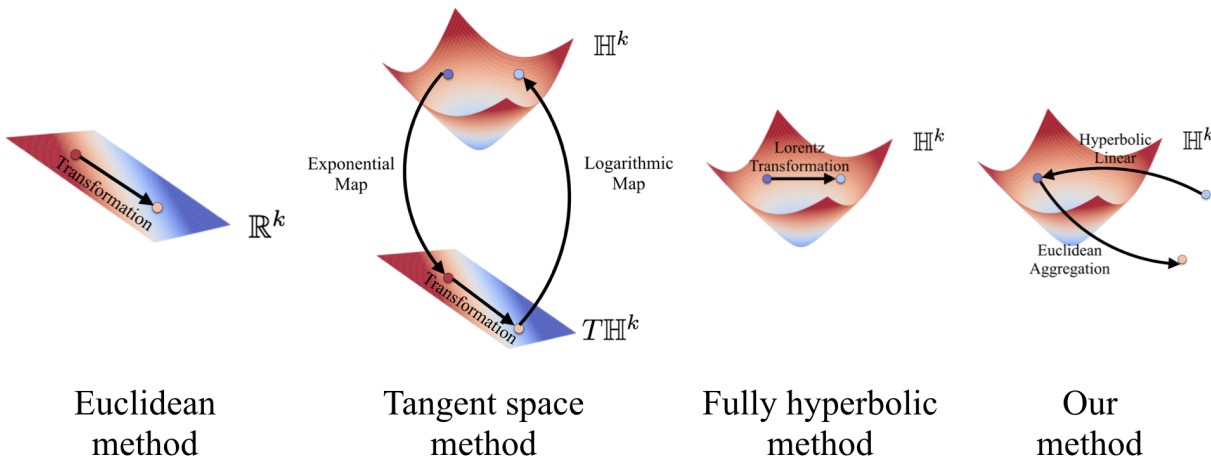

Figure 1: Visual comparison of linear layers for different methods, including the Euclidean method, tangent space method, fully hyperbolic method, and our alternating geometry method.

on $z$ to produce $x^{(1)}$, which serves as input to the subsequent layer. This simplification is possible because our formulation removes the constraint that the input to $HL$ must lie on the manifold. As a result, we can seamlessly integrate hyperbolic and Euclidean layers, offering greater flexibility in network design and potentially capturing both hierarchical and flat structures in the data.

## 3.1 Training the model

After defining the structure of our alternating geometry model, the next crucial step is to determine how to train it effectively. Due to Proposition 8, we note that *any $W$* will result in a valid output on the Lorentz manifold. This property allows us to use regular gradient descent or other standard optimization techniques rather than requiring Riemannian gradient descent. This is a significant contrast to the approach in Chen et al. (2022), where Riemannian gradient descent is necessary. Previous research has shown that hyperbolic embeddings and neural networks can suffer from numerical instabilities (Yu & Sa, 2019; Mishne et al., 2023; Qu & Zou, 2022). These instabilities are often attributed to the use of exponential and logarithmic maps. In our framework, similar to Chen et al. (2022), we avoid these maps in the network parameterization. However, unlike their approach, we also eliminate the need for these maps during optimization. By using Euclidean gradient descent or Adam (Kingma & Ba, 2014), we circumvent the numerical issues that can arise from Riemannian optimization techniques.

## 3.2 Graph Neural Network Architectures

We now define the alternating geometric versions of three different types of GNN architectures: Graph Convolutional Network (GCN) (Kipf & Welling, 2017), GraphConv (GC) (Morris et al., 2019), and Graph Attention Network version 2 (GAT) (Brody et al., 2021). Figure 1 illustrates the difference between existing methods and our method regarding space transformations. We provide the exact update equations below.

**GCN**  A standard GCN layer is given by

$$X^{(\ell+1)} = \hat{L} \cdot X^{(\ell)} \cdot W$$

where $\hat{L}$ is the unnormalized graph Laplacian, $X^{(\ell)}$ are the input features to layer $\ell$, and $W$ is the weight matrix with the learnable parameters.

For the alternating geometry version, we modify the update rule to

$$X^{(\ell+1)} = \hat{L} \cdot HL(X^{(\ell)}; W).$$

Here, $W$ is the weight matrix with learnable parameters, and $HL$ is as defined in Section 3. Note that the feature mapping step uses hyperbolic geometry while the aggregation step (multiplication by the unnormalized Laplacian) occurs in Euclidean space.

**GC** A standard GraphConv layer is given by

$$x_i^{(\ell+1)} = W_1 x_i^{(\ell)} + W_2 \sum_{j \in \mathcal{N}(i)} e_{j,i} x_j^{(\ell)},$$

where $e_{j,i}$ is the edge weight from node $j$ to node $i$, $\mathcal{N}(i)$ is the neighborhood of node $i$, $W_1$ and $W_2$ are matrices with learnable parameters, and $x_i^{(\ell)}$ is the feature vector for the $i$th node.

For the alternating geometry version, we modify the update rule to

$$x_i^{(\ell+1)} = HL(x_i^{(\ell)}; W_1) + HL\left(\sum_{j \in \mathcal{N}(i)} e_{j,i} x_j^{(\ell)}; W_2\right).$$

Again, feature mapping occurs in hyperbolic space while aggregation happens in Euclidean space.

**GAT** A standard Graph Attention layer is given by

$$x_i^{(\ell+1)} = \alpha_{i,i} W x_i^{(\ell)} + \sum_{j \in \mathcal{N}(i)} \alpha_{i,j} W x_j^{(\ell)},$$

where the attention coefficients $\alpha_{i,j}$ are computed as

$$\alpha_{i,j} = \frac{\exp(\text{LeakyReLU}(a^T[W x_i^{(\ell)} \| W x_j^{(\ell)}]))}{\sum_{k \in \mathcal{N}(i) \cup \{i\}} \exp(\text{LeakyReLU}(a^T[W x_i^{(\ell)} \| W x_k^{(\ell)}]))}$$

The matrix $W$ and the vector $a$ are the learnable parameters.

For the alternating geometry version, we modify the update rule to

$$x_i^{(\ell+1)} = \alpha_{i,i} HL(x_i^{(\ell)}; W) + \sum_{j \in \mathcal{N}(i)} \alpha_{i,j} HL(x_j^{(\ell)}; W),$$

where the attention coefficients are now computed as

$$\alpha_{i,j} = \frac{\exp(\text{LeakyReLU}(HL(a^T; [HL(x_i^{(\ell)}; W) \| HL(x_j^{(\ell)}; W)])))}{\sum_{k \in \mathcal{N}(i) \cup \{i\}} \exp(\text{LeakyReLU}(HL(a^T; [HL(x_i^{(\ell)}; W) \| HL(x_k^{(\ell)}; W)])))}.$$

In this case, feature mapping, including the computation of attention weights, uses hyperbolic geometry, while aggregation occurs in Euclidean space.

## 4 Understanding the Effect of Geometry

The primary objective of the paper is to understand the effect of feature geometry on model performance and its interplay with other geometries: downstream task (Loss Geometry), dataset (Data Geometry), and model architecture (Parameter and Function Geometry)[1]. We conduct extensive experimentation to investigate these relationships.

### 4.1 Experimental Setup

We focus on four key aspects that influence generalization error: feature geometry, model architecture, downstream task, and data type. Each unique combination of these factors constitutes an *experimental setting.*

**Downstream Tasks:**   We investigate two distinct tasks: node classification and link prediction. These tasks have different loss functions that result in significantly different loss landscapes.

**Model Architecture:**   We examine three different architectures: Graph Convolutional Networks (GCN), GraphConv (GC), and Graph Attention Networks (GAT). For each architecture, we implement versions corresponding to different geometries.

All our experiments follow a consistent model structure:

1. Input Embedding Layer: For Euclidean GNNs, this is a Euclidean linear layer; for alternating GNNs, it's the hyperbolic linear layer defined in Equation 2; for hyperbolic models, we use their corresponding embedding layer.

2. Message Passing Layers: $k$ layers with width $h$, using ReLU activation.

3. Output Layer:
   - For node classification: An Euclidean linear layer mapping to class labels for all geometries.
   - For link prediction:
     - Euclidean GNNs: Euclidean inner product between node features for each edge.
     - Hyperbolic GNNs: hyperbolic inner product.
     - Alternating geometry GNNs: Euclidean inner product, ignoring the first coordinate (which has special meaning due to the hyperbolic manifold factoring).
     - A sigmoid function is applied to derive edge existence probabilities.

The models are then trained using either Adam or Riemannian Adam with a learning rate of $\eta$.

**Feature Geometry:**   We consider three different feature geometries

1. Euclidean: Standard models - GCN (Kipf & Welling, 2017), GC (Morris et al., 2019), and GAT (Brody et al., 2021).

2. Alternating: Models defined in Section 3.

3. Hyperbolic: LGCN (Zhang et al., 2021b) for GCN, HGNN (Liu et al., 2019) for GC, and HyboNet (Chen et al., 2022) for GAT.

---

[1]All code and data can be found at the Anonymized Github Link

**Datasets:**  We utilize nine diverse datasets spanning different fields and topics:

1. Citation Networks: Cora, Citeseer, PubMed (Yang et al., 2016); nodes represent papers and edges represent citation links.
2. Airport Networks: USA, Brazil, Europe (Ribeiro et al., 2017); nodes denote airports and edges denote air traffic.
3. Website Connection Networks: Cornell, Texas, Wisconsin (Pei et al., 2020b); nodes correspond to web pages and dges correspond to hyperlinks

We note that our consists of two parts the graph and the feature vectors. Each of these components can have their own geometric structure and properties. Table 1 presents the statistics for these datasets, including: node homophily (Zhu et al., 2020), edge homophily (Pei et al., 2020a), mean discrete Olivier Ricci Curvature (ORC) of the graph (Ni et al., 2015), and $\delta$-hyperbolicity of the features (Sonthalia & Gilbert, 2020).

| Dataset | # Nodes | # Edges | Node Homophily | Edge Homophily | ORC | $\delta$ |
|---------|---------|---------|----------------|----------------|------|----------|
| Cora | 2,708 | 10,556 | 0.83 | 0.81 | -0.139 | [0.7, 1.4] |
| Citeseer | 3,327 | 9,104 | 0.71 | 0.74 | 0.030 | [0.8, 1.6] |
| PubMed | 11,717 | 88,648 | 0.79 | 0.80 | -0.286 | |
| USA | 1,190 | 13,599 | 0.34 | 0.70 | 0.096 | 0 |
| Brazil | 131 | 1,038 | 0.28 | 0.47 | 0.125 | 0 |
| Europe | 399 | 5,995 | 0.20 | 0.40 | 0.096 | 0 |
| Cornell | 183 | 298 | 0.11 | 0.13 | -0.060 | 1.94 |
| Texas | 183 | 325 | 0.07 | 0.11 | -0.076 | 1.77 |
| Wisconsin | 251 | 515 | 0.17 | 0.20 | -0.083 | 2.16 |

Table 1: Data Statistics

**Hyperparameter Search:**  For each experimental setting (combination of task, model, geometry, and data), we conduct a search over 36 different hyperparameter configurations:

1. Depths ($k$): 2, 3, 4 (For Pubmed, limited to $k = 2$ due to computational constraints)
2. Widths ($h$): 32, 64, 128, 256
3. Learning rates ($\eta$): 0.0002, 0.001, 0.005

Our hyperparameter search process is as follows:

1. **Data Splitting**: We divide each dataset into train, test, and validation sets.

2. **Training**: For each combination of model, data, and hyperparameters, we perform 10 trials. Each trial consists of 100 epochs for node classification and 200 epochs for link prediction.

3. **Epoch Selection**: For each trial, we select the epoch with the highest validation accuracy as the stopping epoch. In case of ties, we choose the earlier epoch.

4. **Hyperparameter Selection**: We average the validation accuracy across the 10 trials for each hyperparameter configuration. We then pick the model with the best validation error.

## 4.2 Interaction Between Different Geometries

Our analysis focuses on five key geometries that shape the behavior and performance of neural networks: Data Geometry, Parameter Geometry, Function Geometry, Loss Geometry, and Feature Geometry. Each contributes significantly to the model's implicit bias, influencing how the neural network learns and generalizes from data. Their interplay is complex and often subtle, yet fundamentally important.

**Data Geometry** Data Geometry informs us about the structure of the input space. In our study, the datasets remain constant across all models, allowing us to observe how the intrinsic geometry of the data (characterized by ORC and $\delta$-hyperbolicity) affects the results across different model architectures and geometries.

**Parameter Geometry** Parameter Geometry affects optimization dynamics. In our experiments, the Euclidean models, alternating models, and hyperbolic predominantly have parameters in Euclidean space. Only a small number of parameters (at most 10, corresponding to the number of classes) may lie in non-Euclidean space. Given this uniformity, we do not extensively consider the effects of Parameter Geometry in our analysis.

**Function Geometry** Function Space Geometry determines the expressiveness of our model. Each neural network architecture we consider (GCN, GC, GAT) results in a different function space, allowing us to compare how these varying expressivities interact with other geometries. There could be difference in the expressivity for the different geometries.

**Loss Geometry** Loss Geometry guides the learning process and defines the notion of similarity between model outputs. Our study includes two different tasks (node classification and link prediction), each with its distinct loss function. This provides two different loss geometries for comparison. While the parameterization of the neural networks also affects the loss geometry, we have standardized this as much as possible to isolate the effects of the task-specific loss.

**Feature Geometry** Feature Geometry influences the internal representations learned by the network. This is the primary focus of our study, represented by the three different model geometries we consider: Euclidean, hyperbolic, and alternating. By varying this geometry while controlling for other factors, we aim to understand its specific impact on model performance and its interactions with other geometries.

Through our experimental design, we aim to disentangle the effects of these various geometries, with a particular focus on the role of Feature Geometry in graph neural networks.

## 5 Node Classification

We analyze the performance of different model architectures and geometries across various datasets and examine the interplay between model characteristics, dataset properties, and performance metrics. To understand the effect of geometry on performance, we first analyze the best hyperparameter configuration for each model for each dataset. Table 2 presents these results. For the distribution of the accuracies with respect to the hyperparameters can be seen in the Appendix A.3.

### 5.1 Function Geometry and Feature Geometry Interaction

We first consider the interaction of function geometry (the model architecture) and feature geometry. We compute the mean accuracies for each model architecture and feature geometry combination to do this. This can be seen in Table 3.

| Dataset | Euclidean | | | Alternating | | | Hyperbolic | | |
|---|---|---|---|---|---|---|---|---|---|
| | GCN | GC | GAT | GCN | GC | GAT | GCN | GC | GAT |
| Airports Brazil | $24 \pm 1$ | $42 \pm 2$ | $43 \pm 1$ | $20 \pm 1$ | $39 \pm 1$ | $41 \pm 1$ | $30 \pm 3$ | $40 \pm 3$ | $17 \pm 0$ |
| Airports Europe | $43 \pm 3$ | $42 \pm 1$ | $41 \pm 1$ | $46 \pm 1$ | $47 \pm 1$ | $42 \pm 1$ | $44 \pm 0$ | $49 \pm 1$ | $40 \pm 0$ |
| Airports USA | $46 \pm 0$ | $52 \pm 0$ | $46 \pm 1$ | $47 \pm 0$ | $49 \pm 0$ | $45 \pm 1$ | $47 \pm 0$ | $49 \pm 1$ | $45 \pm 0$ |
| Citeseer | $65 \pm 0$ | $64 \pm 0$ | $65 \pm 0$ | $66 \pm 0$ | $66 \pm 0$ | $64 \pm 1$ | $68 \pm 0$ | $68 \pm 0$ | $65 \pm 1$ |
| Cora | $79 \pm 0$ | $78 \pm 0$ | $79 \pm 0$ | $78 \pm 0$ | $80 \pm 0$ | $78 \pm 0$ | $79 \pm 0$ | $79 \pm 0$ | $80 \pm 0$ |
| Pubmed | $74 \pm 1$ | $77 \pm 0$ | $74 \pm 0$ | $75 \pm 0$ | $74 \pm 0$ | $74 \pm 0$ | $76 \pm 0$ | $77 \pm 0$ | $74 \pm 0$ |
| Cornell | $45 \pm 1$ | $52 \pm 1$ | $48 \pm 1$ | $47 \pm 1$ | $45 \pm 1$ | $46 \pm 1$ | $43 \pm 1$ | $43 \pm 1$ | $44 \pm 0$ |
| Texas | $61 \pm 0$ | $67 \pm 1$ | $61 \pm 1$ | $65 \pm 1$ | $65 \pm 2$ | $64 \pm 1$ | $56 \pm 1$ | $51 \pm 1$ | $56 \pm 1$ |
| Wisconsin | $51 \pm 0$ | $62 \pm 0$ | $51 \pm 1$ | $52 \pm 0$ | $69 \pm 1$ | $55 \pm 0$ | $51 \pm 0$ | $52 \pm 0$ | $48 \pm 1$ |

Table 2: Average accuracy of the best hyperparameter configuration for each model on each dataset along with the standard deviation. Each entry is the average of ten trials. The darker cell color refers to better accuracy.

| | Euclidean | Alternating | Hyperbolic | Mean |
|---|---|---|---|---|
| GCN | $54.33 \pm 1.76$ | $\underline{55.11 \pm 1.81}$ | $55.06 \pm 1.67$ | $54.83 \pm 1.01$ |
| GC | $\mathbf{\underline{59.35 \pm 1.4}}$ | $\mathbf{59.34 \pm 1.47}$ | $\mathbf{56.44 \pm 1.48}$ | $\mathbf{58.38 \pm 0.84}$ |
| GAT | $56.27 \pm 1.41$ | $\underline{56.69 \pm 1.44}$ | $52.15 \pm 1.93$ | $55.04 \pm 0.94$ |
| Mean | $56.65 \pm 0.89$ | $\underline{57.05 \pm 0.92}$ | $54.55 \pm 0.99$ | |

Table 3: Average accuracy for the best hyperparameter configuration for each architecture - feature geometry pair and the standard error. We also aggregate the average for each row (architecture) and column (feature geometry). The best number in each row is underlined, and the best in each column is in bold.

We consider the performance of the different architectures for each feature geometry and see that the GraphConv (GC) model has the best performance for each of the different feature geometries. Hence, we observe that on aggregate, GraphConv (GC) models outperform Graph Convolutional Neural Network (GCN) and Graph Attention Network (GAT) models by 3.45% and 3.34%, respectively. Next, we consider the performance of each of the different feature geometries for each model architecture. Here, we see that the Euclidean and alternating models have the best performance and outperform the hyperbolic models by 2.15% and 2.5%, respectively.

Additionally, the mean accuracy for the hyperbolic GC models is higher than the accuracies for Euclidean GCN, GAT, and alternating GCN. Hence, this suggests that architecture has a more significant impact than feature geometry for node classification. However, with the overlapping uncertainty intervals for the accuracy, we cannot definitively claim that the hyperbolic GC outperforms those models.

## 5.2 Data Geometry and Feature Geometry Interaction

Hyperbolic neural networks have recently become quite prominent. One reason attributed to their success is that many different types of data are naturally hyperbolic. Our datasets consist of two different components - the graph and the feature vectors. For the graph component, the average

ORC measures the curvature of the graph. We split our datasets into those with negative (Cora, PubMed, Cornell, Texas, Wisconsin) and positive curvature (Citeseer, USA, Brazil, Europe). We tabulate the average accuracy across architectures for each geometry and plot our results in Figure 2. For datasets with positive curvature, we see that accuracies across geometry fall within a range of 0.8%, with alternating geometry models yielding the highest average accuracy. This deviation is within the standard error of the means. Hence, the different feature geometries result in similar performance. However, for datasets with negative curvature, we see that accuracies for hyperbolic models are 3.3 % and 3.9% lower than for Euclidean and alternating models respectively.

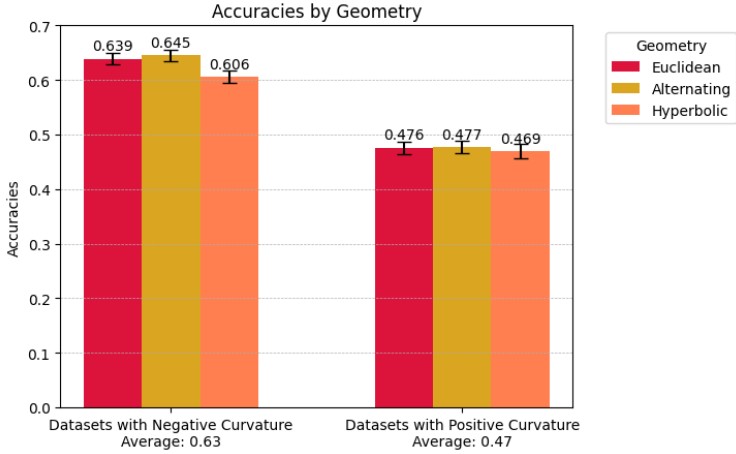

Figure 2: Model accuracies of best configurations by geometry, averaged over datasets with negative and positive curvature

We also have the geometry of the input features. We use the Gromov hyperbolicity $\delta$ to measure this. The Gromov hyperbolicity tells us how hyperbolic a dataset is. The smaller the value the more hyperbolic. Here, we divide the datasets into datasets with small $\delta$ (USA, Brazil, and Europe) vs the datasets with large $\delta$ (Wisconsin, Texas, Cornell). The results are shown in Figure 3. Here, we see that for datasets with small $\delta$, we do not see much difference between the different feature geometries. However, we see that using hyperbolic features degrades performance significantly for datasets with large $\delta$.

The belief used to be that hyperbolic models perform well on hyperbolic data. However, we see that negatively curved graphs seem to hinder the performance of hyperbolic models. On the other hand, we see much better synergy between input feature geometry and network feature geometry. *Specifically, if the input geometry is not hyperbolic, then the model suffers.*

### 5.3 Other Data Properties and Model Geometry

Other factors could cause the trends we see above. We explore two other factors here - the size of the data set and the node homophily.

**Dataset Size** To understand the impact of the size of the graph on model performance, we divide our datasets into small and large, as defined by the number of edges. We consider a dataset small if it has less than 1038 edges and a datasets large if it has more than 5995 edges. Thus, Airports-Brazil, WebKB-Cornell, WebKB-Texas, and WebKB-Wisconsin are small, and Cora, Citeseer, PubMed, Airports-Europe, and Airports-USA are large. Figure 4 displays the accuracies for the small and large datasets. We see that for large datasets the different feature geometries result in the same

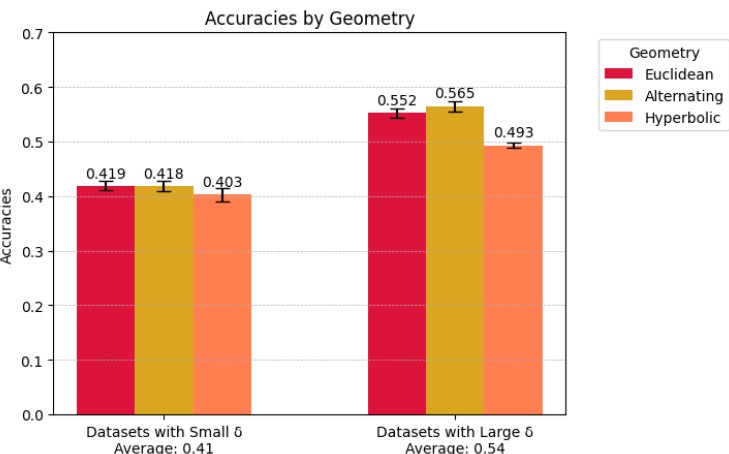

Figure 3: Model accuracies of best configurations by geometry, averaged over datasets with large and small $\delta$

overall performance, within 1.2%. However, for small datasets, we see that hyperbolic models underperform compared to alternating and Euclidean geometry models by a difference of 6.6% and 6.2% respectively, suggesting that hyperbolic models suffer the most from smaller datasets.

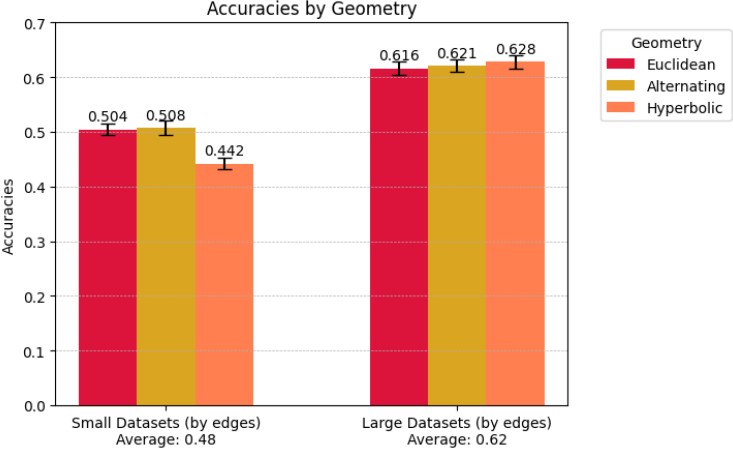

Figure 4: Model accuracies of best configurations by geometry, averaged over small and large datasets, as categorized by the number of edges

**Node Homophily** Out of our 9 datasets, we see that we are able to separate our datasets into those with high node homophily and low node homophily. These are:

- Low Node Homophily: Airports-Brazil (0.34), Airports-Europe (0.28), Airports-USA (0.20), WebKB-Cornell (0.11), WebKB-Texas (0.07), WebKB-Wisconsin (0.17)

- High Node Homophily: Cora (0.83), Citeseer (0.71), Pubmed (0.79)

We look at the performance of each feature geometry between datasets with low and high node homophily and report our findings in Figure 5. On datasets with high node homophily, we see that accuracies across feature geometries fall within a range of 1.2%. However, on low datasets with low

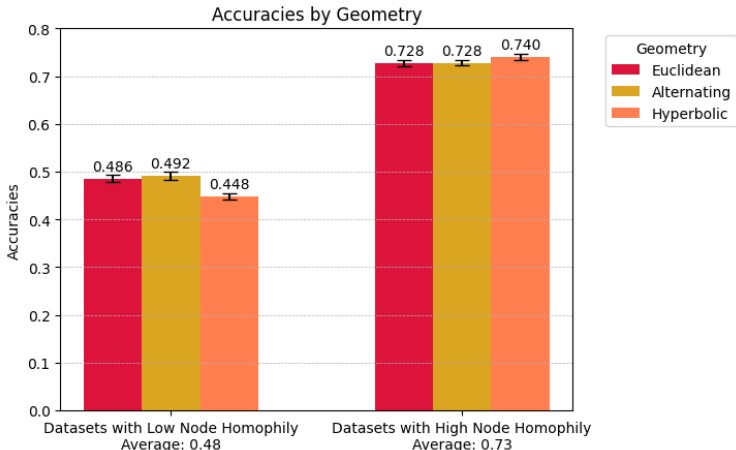

Figure 5: Model accuracies of best configurations by geometry, aggregated over datasets with low and high node homophily

node homophily, we see that hyperbolic models significantly underperform compared to Euclidean and alternating geometry models by a margin of 3.8% and 4.4% respectively.

However, both of these results could be due to the fact that our datasets with low node homophily and small datasets contain our datasets with large Gromov hyperbolicity, which may be the important factor.

### 5.4 Best Hyperparameters

In this section, we look closely at the hyperparameters of the best configurations for each model on each dataset. We then aggregate over geometry and then architecture to study trends in the preferences that our architectures and geometries have to each hyperparameter. We plot our results in Figures 6 and 7.

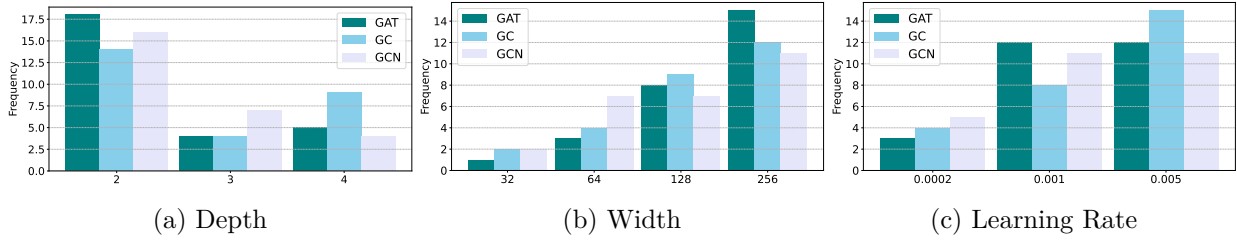

(a) Depth        (b) Width        (c) Learning Rate

Figure 6: Figure showing the distribution of the best hyperparameters for each architecture type on node classification experiments.

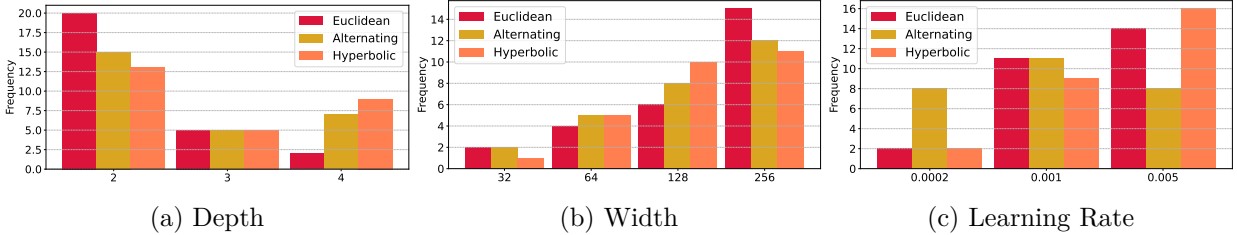

Figure 7: Figure showing the distribution of the best hyperparameters for each geometry type on node classification experiments.

We observe that all three architectures yield higher accuracies with wider models, with GAT showing the greatest sensitivity to the number of dimensions. We also note that each of the architectures perform better on shallower models, along with higher learning rates. Notably, GC seems to be the least sensitive to the number of layers but exhibits the highest sensitivity to the learning rate.

Examining the aggregated distribution over geometry, we observe a consistent preference for wider and shallower networks. Among the geometries, Euclidean models exhibit the highest sensitivity to changes in hyperparameters. In terms of learning rate, both Euclidean and hyperbolic models favor higher rates, whereas alternating models appear largely indifferent. Similar to the trends observed with architecture, all geometries perform better with shallower networks. However, alternating and hyperbolic models are notably less sensitive to the number of layers compared to their Euclidean counterparts.

### 5.4.1 Sensitivity to Hyperparameter

We next compare the aggregates of the best configuration per combination of model and dataset with the aggregates over all configurations. This provides us an understanding of the stability of accuracy across hyperparameters for each architecture and geometry. The results are summarized below in Table 4. We see that out of the three architectures, GC benefits the most from hyperparameter tuning, with an improvement of 5.58% between best and all configurations as compared to the improvements of 4.21% and 3.16% for GCN and GAT respectively. Taking a look at geometry, we see that alternating models are least sensitive to hyperparameters, with an improvement in accuracy between best and all configurations of 3.24%, compared to the higher margins of improvement of 4.58% and 5.14% of Euclidean and hyperbolic respectively.

### 5.5 Takeaways for Node Classification

- On average, we see that GC is the best performing architecture, while Euclidean and alternating models make up the best performing geometries. We observe that choice of architecture potentially has a more significant impact than choice of geometry on the accuracy of the model

- Hyperbolic models perform poorly on dataset whose features are not hyperbolic

- When node homophily of the dataset is low, GC outperforms other architectures. When the Node homophily is sufficiently large, there is insignificant difference between the models

- There is a consistent preference for shallower and wider configurations across all models. We further observe that the GC architecture is the most sensitive to hyperparameter selec-

|        | Euclidean         | Alternating          | Hyperbolic        | Mean                 |
|--------|-------------------|----------------------|-------------------|----------------------|
| GCN    | $54.33 \pm 1.76$  | $\underline{55.11 \pm 1.81}$ | $55.06 \pm 1.67$  | $54.83 \pm 1.01$     |
| GC     | $\mathbf{59.35 \pm 1.4}$ | $\mathbf{59.34 \pm 1.47}$ | $\mathbf{56.44 \pm 1.48}$ | $\mathbf{58.38 \pm 0.84}$ |
| GAT    | $56.27 \pm 1.41$  | $\underline{56.69 \pm 1.44}$ | $52.15 \pm 1.93$  | $55.04 \pm 0.94$     |
| Mean   | $56.65 \pm 0.89$  | $\underline{57.05 \pm 0.92}$ | $54.55 \pm 0.99$  |                      |

(a) Best configurations

|        | Euclidean         | Alternating          | Hyperbolic        | Mean                 |
|--------|-------------------|----------------------|-------------------|----------------------|
| GCN    | $49.07 \pm 0.33$  | $\underline{52.19 \pm 0.31}$ | $\mathbf{50.59 \pm 0.28}$ | $50.62 \pm 0.18$     |
| GC     | $\mathbf{54.95 \pm 0.24}$ | $\mathbf{\underline{55.74 \pm 0.26}}$ | $47.71 \pm 0.27$  | $\mathbf{52.8 \pm 0.16}$ |
| GAT    | $52.2 \pm 0.27$   | $\underline{53.5 \pm 0.24}$  | $49.94 \pm 0.29$  | $51.88 \pm 0.19$     |
| Mean   | $52.07 \pm 0.18$  | $\underline{53.81 \pm 0.17}$ | $49.41 \pm 0.17$  |                      |

(b) All configurations

Table 4: For these tables, we aggregate the accuracies by models over all 9 datasets. In (a), we consider only the best configuration for each model x dataset, whereas for (b), we average over all configurations for each model x dataset

tion, while, amongst the geometries, alternating models show the greatest stability across hyperparameters.

# 6 Link Prediction

We now conduct a similar analysis on the data from our link prediction experiments. As with node classification, we assess differences in AUC with varying graph properties over geometry and architecture. We dive deep into some of our key findings within the main paper, with other findings listed in Appendix B.

## 6.1 Overview of Best AUC scores

Once again, we tabulate the performance for the best hyperparameter configuration for each model on each dataset. We also tabulate the averages over geometry and architecture. These results can be found in Tables 5 and 6 respectively.

We observe that alternating geometry models have lower AUC on most experiments. We also see outliers, such as Euclidean GCN on Cornell and hyperbolic GC on Texas, which have lower AUCs than other models on the same datasets. As we use the same procedure for all our experiments, we report our analysis while acknowledging the possible limitations caused by the outlier experiments.

From Table 5 we see that all the hyperbolic models have AUC scores of about 99% for the Publications datasets. Examining the results from Table 6, we see that GC outperforms the other architectures by an average difference of 3.13%, with GCN and GAT achieving similar AUCs. When looking at geometry, we see that hyperbolic outperforms alternating and Euclidean by a margin of 16.13% and 2.93% respectively. We do see the GC models produce the highest AUCs for Euclidean and alternating but not for hyperbolic. Conversely, the hyperbolic models achieve the

| Dataset | Euclidean | | | Alternating | | | Hyperbolic | | |
|---|---|---|---|---|---|---|---|---|---|
| | GCN | GC | GAT | GCN | GC | GAT | GCN | GC | GAT |
| Airports Brazil | $84 \pm 0$ | $86 \pm 0$ | $84 \pm 0$ | $70 \pm 4$ | $83 \pm 1$ | $71 \pm 2$ | $93 \pm 0$ | $84 \pm 0$ | $91 \pm 0$ |
| Airports Europe | $84 \pm 0$ | $85 \pm 0$ | $82 \pm 0$ | $66 \pm 3$ | $84 \pm 0$ | $77 \pm 1$ | $91 \pm 0$ | $83 \pm 0$ | $87 \pm 0$ |
| Airports USA | $91 \pm 0$ | $93 \pm 0$ | $87 \pm 0$ | $73 \pm 4$ | $92 \pm 0$ | $73 \pm 2$ | $96 \pm 0$ | $93 \pm 0$ | $95 \pm 0$ |
| Citeseer | $94 \pm 0$ | $96 \pm 0$ | $89 \pm 0$ | $74 \pm 3$ | $79 \pm 1$ | $75 \pm 2$ | $99 \pm 0$ | $99 \pm 0$ | $99 \pm 0$ |
| Cora | $96 \pm 0$ | $94 \pm 0$ | $94 \pm 0$ | $72 \pm 3$ | $79 \pm 1$ | $71 \pm 3$ | $99 \pm 0$ | $98 \pm 0$ | $99 \pm 0$ |
| Pubmed | $93 \pm 0$ | $94 \pm 0$ | $88 \pm 0$ | $80 \pm 2$ | $87 \pm 1$ | $75 \pm 2$ | $99 \pm 0$ | $96 \pm 0$ | $99 \pm 0$ |
| Cornell | $68 \pm 1$ | $82 \pm 1$ | $82 \pm 1$ | $60 \pm 3$ | $84 \pm 1$ | $57 \pm 1$ | $81 \pm 1$ | $62 \pm 1$ | $88 \pm 1$ |
| Texas | $74 \pm 1$ | $78 \pm 1$ | $75 \pm 1$ | $61 \pm 3$ | $73 \pm 1$ | $58 \pm 2$ | $75 \pm 1$ | $58 \pm 1$ | $79 \pm 1$ |
| Wisconsin | $83 \pm 0$ | $81 \pm 1$ | $75 \pm 0$ | $52 \pm 5$ | $74 \pm 1$ | $55 \pm 2$ | $80 \pm 1$ | $78 \pm 1$ | $86 \pm 0$ |

Table 5: Average AUC of the best configuration for each model on each dataset along with the standard deviation for link prediction. Each entry is with respect to ten trials. The darker the cell color indicates better AUCs.

| | Euclidean | Alternating | Hyperbolic | Mean |
|---|---|---|---|---|
| GCN | $85.15 \pm 0.96$ | $67.62 \pm 1.34$ | $\underline{90.25 \pm 0.95}$ | $81.01 \pm 0.87$ |
| GC | $\underline{\mathbf{87.62 \pm 0.68}}$ | $\mathbf{81.46 \pm 0.65}$ | $83.69 \pm 1.51$ | $\mathbf{84.26 \pm 0.61}$ |
| GAT | $83.91 \pm 0.67$ | $68.01 \pm 1.11$ | $\underline{\mathbf{91.52 \pm 0.73}}$ | $81.15 \pm 0.77$ |
| Mean | $85.56 \pm 0.46$ | $72.36 \pm 0.73$ | $\underline{88.49 \pm 0.67}$ | |

Table 6: Average AUC for the best hyperparameter configuration for each architecture-geometry pair and the standard error. We also aggregate the average for each row (architecture) and column (geometry). The best number in each row is underlined, and the best in each column is in bold.

highest results for both GCN and GAT but not for GC. We also note that alternating models yield comparable results for GC models but struggle with GCN and GAT architectures.

## 6.2 Function Geometry and Feature Geometry Interaction

To understand the interaction between function geometry (model architecture) and feature geometry, we analyze the mean AUCs for each combination. Table 6 presents these results.

Examining the performance across architectures for each feature geometry, we observe that the GraphConv (GC) models consistently outperform Graph Convolutional Neural Network (GCN) and Graph Attention Network (GAT) models. Specifically, GC models achieve an average AUC of 84.26%, surpassing GCN and GAT by 3.25% and 3.11% respectively. However, this advantage is primarily driven by the strong performance of alternating GC models, which significantly outperform their GCN and GAT counterparts. Focusing on the Euclidean and hyperbolic models, we see that GC models no longer hold an advantage and average an AUC of approximately 2.05% less than GCN and GAT models, which exhibit nearly identical performance.

When considering the performance of different feature geometries for each architecture, we notice a striking trend. Hyperbolic geometry consistently yields the best performance, with the exception of GC models. For GCN, hyperbolic geometry outperforms Euclidean and alternating geometries by 5.1% and 22.63% respectively. Similarly for GAT, hyperbolic geometry shows improvements

of 7.61% and 23.51% over Euclidean and alternating geometries. Interestingly, for GC models, Euclidean geometry slightly edges out hyperbolic geometry by 3.93%, while still significantly out-performing alternating geometry by 6.16%. This suggests that the choice of geometry has a more nuanced effect on GC models compared to GCN and GAT.

The superior performance of hyperbolic models is particularly evident in GCN and GAT archi-tectures, where they achieve the highest AUCs of 90.25% and 91.52% respectively. These scores are notably higher than the best performance of GC models (87.62% with Euclidean geometry), indicating that for link prediction, the combination of hyperbolic geometry with GCN or GAT architectures is particularly effective.

These observations lead us to conclude that *the choice of geometry is more significant than the choice of architecture for the link prediction task.* The consistent superiority of hyperbolic models, especially when paired with GCN or GAT architectures, suggests that the hyperbolic space is particularly well-suited for capturing the structural information needed for link prediction.

## 6.3 Data Geometry and Feature Geometry Interaction

We next analyze the interaction between model geometry and graph curvature for the link prediction task. As in the node classification analysis of section 5.2, we partition the datasets into those with negative and positive curvature and average the AUC scores of the best configurations across geometries. The results are shown in Figure 8.

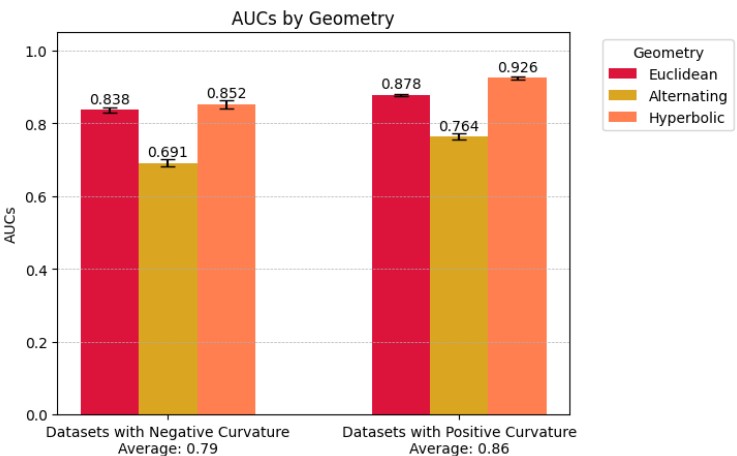

Figure 8: AUC scores of best configurations by geometry, aggregated over datasets with negative and positive curvature

For datasets with negative curvature, we note that hyperbolic models outperform the Euclidean models by 1.4%, as compared to the larger margin of 4.8% on datasets with positive curvature. These findings align with our observations from node classification, where graphs with negative curvature appeared to hinder the relative performance of hyperbolic models. Notably, models with alternating geometry perform significantly worse on datasets with both positive and negative curvature, with an average AUC of about 14% less than the other geometries, consistent across both sets of datasets.

We further examine the effects of input geometry on link prediction performance by dividing our datasets based on Gromov hyperbolicity $\delta$, as was done in the Node Classification section. Figure

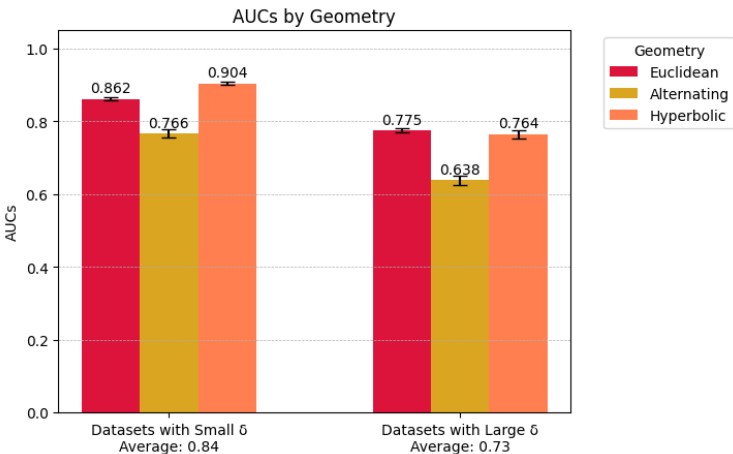

Figure 9: AUC scores of best configurations by architecture, aggregated over datasets with large and small hyperbolicity

9 presents the relative accuracies of the best configurations aggregated by geometry and split by hyperbolicity. For datasets with small hyperbolicity, we observe that hyperbolic models achieve the highest average AUC of 90.4%, outperforming Euclidean and alternating geometry models by 4.2% and 13.8%, respectively. In contrast, for datasets with large hyperbolicity, overall performance decreases across all geometries. These results suggest that hyperbolic models are most effective when the input features exhibit low hyperbolicity. Conversely, in datasets with high hyperbolicity, hyperbolic models lose their advantage and fail to outperform Euclidean models.

### 6.4 Other Data Properties and Model Geometry

**Dataset Size**   To explore the impact of dataset size on model performance, we categorize the datasets into small and large based on the number of edges, consistent with our node classification analysis in section 5.3. To recall, our small datasets are Airports-Brazil, WebKB-Cornell, WebKB-Texas, and WebKB-Wisconsin, whereas Cora, Citeseer, PubMed, Airports-Europe, and Airports-USA constitute our large datasets. Table 7 and Figure 10 summarize our results.

| | Small Datasets | | | Large Datasets | | |
|---|---|---|---|---|---|---|
| Model | Hyperbolic | Euclidean | Alternating | Hyperbolic | Euclidean | Alternating |
| GAT | $86.0 \pm 0.8$ | $79.0 \pm 0.8$ | $60.4 \pm 1.4$ | $95.9 \pm 0.7$ | $87.9 \pm 0.6$ | $74.1 \pm 1.1$ |
| GC | $70.8 \pm 1.8$ | $81.8 \pm 0.6$ | $78.3 \pm 0.9$ | $95.0 \pm 0.5$ | $92.3 \pm 0.6$ | $84.0 \pm 0.9$ |
| GCN | $82.1 \pm 1.1$ | $77.3 \pm 1.2$ | $61.0 \pm 2.1$ | $96.8 \pm 0.5$ | $91.4 \pm 0.7$ | $72.9 \pm 1.5$ |

Table 7: Hyperbolic vs Euclidean Models by dataset size

From Figure 10, we observe that on small datasets, hyperbolic and Euclidean models achieve similar performance, averaging around 79.5% AUC, with alternating models continuing to underperform significantly at an average AUC of 66.6%. On large datasets, hyperbolic models demonstrate a clear advantage, achieving an average AUC that is 5.1% higher than Euclidean models, while alternating models lag further behind.

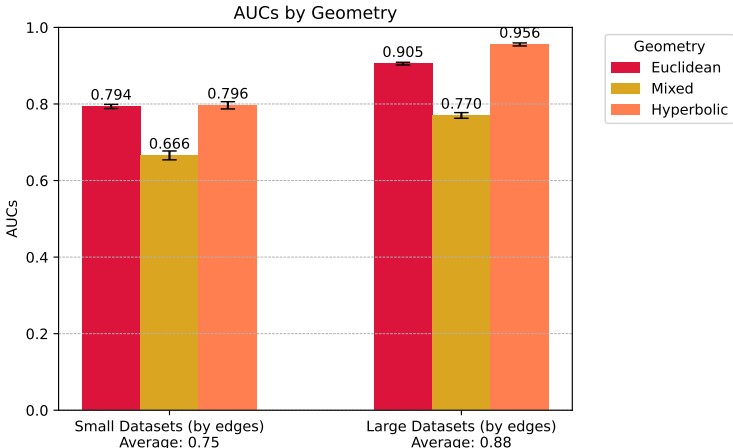

Figure 10: AUC scores of best configurations by architecture, aggregated over small and large datasets

A closer inspection of Table 7 reveals that the performance advantage of hyperbolic models on large datasets over Euclidean models is consistent across all architectures. On small datasets on the other hand, while hyperbolic models generally outperform Euclidean models, an exception is observed with GC models, where the Euclidean GC outperforms the hyperbolic GC by 10%. Referring back to Table 5, we see that hyperbolic GC particularly struggles on the Cornell and Texas datasets, producing an average AUC of 20% less than the Euclidean GC model. Notably, these are the two smallest datasets we observe, indicating the hyperbolic GC's sensitivity to graph size.

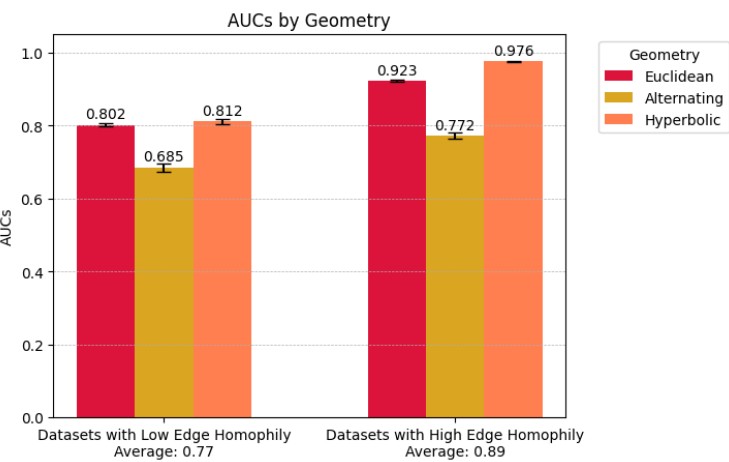

Figure 11: AUC scores of best configurations by architecture, aggregated over datasets with high and low edge homophily

**Edge Homophily**  In this section, we evaluate the impact of edge homophily on model performance. Out of our 9 datasets, we are able to separate them into those with high edge homophily and low edge homophily. These are:

- Low Edge Homophily: WebKB-Cornell (0.13), WebKB-Texas (0.11), WebKB-Wisconsin (0.20), Airports-Brazil (0.47), Airports-Europe (0.40)

- High Edge Homophily: Airports-USA (0.70), Cora (0.81), Citeseer (0.74), Pubmed (0.80)

We average the performance of each feature geometry across datasets with low and high edge homophily, reporting our findings in Figure 11. We see that on datasets with high edge homophily, hyperbolic models outperform their Euclidean counterparts by a margin of 5.3%. However, this relative advantage is diminished amongst datasets with low edge homophily, where this margin is reduced to 1%. However, we note that these results could be due to the fact that there is significant overlap between our small datasets (as defined by number of edges) and those with low edge homophily, which make it hard to identify the extent of impact the size and edge homophily have on model performance.

## 6.5 Best Hyperparameters

In this section, we examine the hyperparameters of the best configurations for each model in each data set. As with node classification, we aggregate the results first by geometry and then by architecture to identify trends in hyperparameter preferences. The results are visualized in Figures 12 and 13.

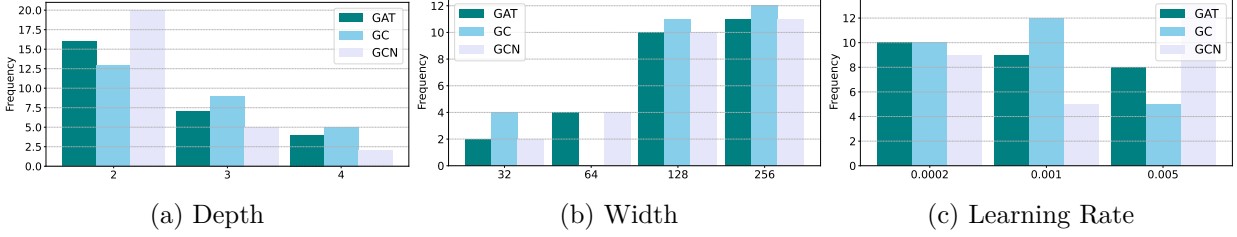

(a) Depth         (b) Width         (c) Learning Rate

Figure 12: Figure showing the distribution of the best hyperparameters for each architecture type.

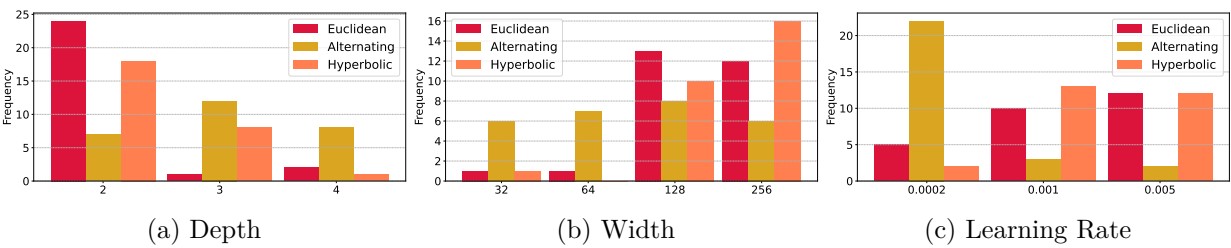

(a) Depth         (b) Width         (c) Learning Rate

Figure 13: Figure showing the distribution of the best hyperparameters for each geometry type.

We observe a consistent preference across architectures for shallower networks, with higher AUC scores achieved on models with fewer layers. Among the architectures, GCN exhibits the strongest sensitivity to depth—out of the 27 models tested (9 datasets × 3 geometries), GCN's best configurations favor two layers in 19 cases. Additionally, all architectures show a strong preference for wider networks, mirroring trends observed in node classification. However, preferences for learning rates are less clear, with no consistent pattern emerging across architectures.

Aggregating results by geometry, we find that Euclidean and hyperbolic models exhibit a strong preference for wider and shallower networks. Alternating models, in contrast, display a notable preference for the smallest learning rate, while Euclidean and hyperbolic models tend to favor the largest learning rates. Consistent with architecture-level trends, all geometries perform better with

shallower networks, although Euclidean models show slightly greater sensitivity to the number of layers compared to hyperbolic and alternating models.

### 6.5.1 Sensitivity to Hyperparameter

|      | Euclidean | Alternating | Hyperbolic | Mean |
|------|-----------|-------------|------------|------|
| GCN  | $85.15 \pm 0.96$ | $67.62 \pm 1.34$ | $\underline{90.25 \pm 0.95}$ | $81.01 \pm 0.87$ |
| GC   | $\mathbf{87.62 \pm 0.68}$ | $\mathbf{81.46 \pm 0.65}$ | $83.69 \pm 1.51$ | $\mathbf{84.26 \pm 0.61}$ |
| GAT  | $83.91 \pm 0.67$ | $68.01 \pm 1.11$ | $\mathbf{\underline{91.52 \pm 0.73}}$ | $81.15 \pm 0.77$ |
| Mean | $85.56 \pm 0.46$ | $72.36 \pm 0.73$ | $\underline{88.49 \pm 0.67}$ | |

(a) Best configuration

|      | Euclidean | Alternating | Hyperbolic | Mean |
|------|-----------|-------------|------------|------|
| GCN  | $79.23 \pm 0.18$ | $57.37 \pm 0.17$ | $\underline{88.04 \pm 0.18}$ | $74.88 \pm 0.22$ |
| GC   | $\mathbf{81.57 \pm 0.14}$ | $\mathbf{78.64 \pm 0.14}$ | $\underline{81.96 \pm 0.27}$ | $\mathbf{80.72 \pm 0.22}$ |
| GAT  | $79.68 \pm 0.16$ | $58.0 \pm 0.17$ | $\mathbf{\underline{90.66 \pm 0.13}}$ | $76.11 \pm 0.23$ |
| Mean | $80.16 \pm 0.18$ | $64.67 \pm 0.21$ | $\underline{86.89 \pm 0.19}$ | |

(b) All configurations

Table 8: For these tables, we aggregate the AUC by models over all 9 datasets. In (a) we consider only the best configuration for each model x dataset, whereas for (b) we average over all configurations for each model x dataset.

We next compare the aggregates of the best configuration for each model-dataset combination with the aggregates over all configurations to assess the stability of AUC across hyperparameters. The results are summarized in Table 8.

Examining the architectures, we find that GC is the least sensitive to hyperparameters, with an improvement of 3.54% between all configurations and the best configuration results. In contrast, GCN and GAT show greater sensitivity, with improvements of 6.19% and 5.05%, respectively. This indicates that GC performs consistently across a wider range of hyperparameter settings, while GCN and GAT rely more heavily on fine-tuned configurations.

Analyzing the geometries, we observe that hyperbolic models exhibit the smallest improvement of 1.6%, suggesting they are the least sensitive to hyperparameter tuning. In comparison, Euclidean and alternating models show more pronounced improvements of 5.4% and 7.6%, respectively, highlighting their higher dependence on optimal hyperparameter selection. Alternating models, while showing the greatest sensitivity overall, display less improvement in specific combinations such as GC, indicating potential stability in certain scenarios.

From the distributions shown in Appendix B.3, we see that hyperbolic models exhibit low variance in performance across configurations, whereas Euclidean and alternating models exhibit broader variability. This aligns with the observed improvements and supports the conclusion that hyperbolic models are the most robust to hyperparameter variations.

## 6.6 Takeaways for Link Prediction

To summarize our findings, we see that:

- Hyperbolic models have the highest AUC compared to Euclidean and alternating models. The latter struggles significantly on the link prediction task. We also note that the GC has the highest average AUC out of the architectures explored.

- The choice of geometry is more significant than the choice of architecture for the performance of the model.

- Hyperbolic models have an advantage over Euclidean models on datasets with small hyperbolicity.

- While hyperbolic models maintain an advantage over Euclidean models generally, the hyperbolic GC is specifically sensitive to the size of the dataset, performing significantly worse than the Euclidean GC on small datasets.

- We see that there is a general preferences amongst all models for shallower and wider networks, with inconclusive preferences for learning rates. Hyperbolic models exhibit the lowest sensitivity to hyperparameter selection along with the greatest stability across trials[2].

## 7 Meta Analysis

Comparing the results from node classification and link prediction tasks reveals several interesting insights about the interplay between geometry, architecture, and task-specific performance.

## 7.1 Task-Agnostic Observations

**Model Depth and Width:** Across both tasks, shallower and wider models tend to perform better for Euclidean and hyperbolic geometries. Alternating geometry models show more flexibility regarding depth.

**GraphConv (GC) Performance:** GC consistently demonstrates strong performance across both tasks, often outperforming other architectures regardless of geometry.

**Euclidean Baseline:** Euclidean models consistently provide solid performance, ranking second-best for both tasks. This suggests that Euclidean models serve as a reliable baseline for comparison.

**Hyperparameter Sensitivity:** The best-performing models for each task tend to be less sensitive to hyperparameter choices and initialization, indicating a correlation between performance and stability.

**Hyperbolic models perform better on Positively Curved Graphs:** Node Negatively curved graphs hinder hyperbolic model performance for node classification. For link prediction, hyperbolic models perform well on both positively and negatively curved graphs, with a slight advantage on positively curved graphs.

---

[2]See Appendix B.4 for an exploration of stability across trials.

## 7.2 Task-Dependent Observations

While several high-level insights hold across tasks, our analysis reveals distinct trends that highlight the more nuanced relationship between geometry, architecture, and task.

**Best Geometry-Architecture Combination:**

- **Node Classifcation**: The optimal combination involves alternating geometry with Graph-Conv (GC) - here, GC and alternating geometry are the highest performing architecture and geometry respectively.

- **Link Prediction**: Hyperbolic geometry with Graph Attention Networks (GAT) yielded the highest accuracies, despite GC being the best-performing architecture overall. Here, hyperbolic models outperformed other geometries by a notable margin.

**Impact of Geometry vs Architecture:**

- **Node Classification**: We see that architecture has a more significant impact on model accuracy than geometry, although the performance differences between geometries and architectures remain relatively small.

- **Link Prediction**: We see that geometry plays a more dominant role in influencing model performance than architecture, with substantial differences in AUC between geometries.

## 8 Conclusion and Future Work

**Task-Specific Geometry Selection:** Our results emphasize the importance of choosing the appropriate geometry based on the specific task at hand. Future work could explore automated methods for selecting the optimal geometry given a particular dataset and task.

**Hybrid Models:** The contrasting performance of alternating geometry models between tasks suggests potential for hybrid approaches that leverage the strengths of different geometries for different components of the learning process.

**Architecture-Geometry Synergy:** The varying impact of architecture and geometry choices across tasks highlights the need for a deeper understanding of how these elements interact. For instance, we see that in Table 6, hyperbolic GCN and GAT models yield extremely strong performance, but appear to underperform when paired with the GC architecture, despite the fact that GC models perform the best overall. This paves the way for further research into the theoretical foundations of interactions between geometry and architecture.

**Dataset Characteristics:** The observed differences in performance based on dataset size and graph curvature indicate that these factors should be considered when designing or selecting GNN models. For example, in our work, we observed that hyperbolic models were not suitable for small datasets for the node classification task. Developing models that can adapt to such characteristics could lead to more robust performance across diverse datasets.

**Stability and Performance:** The correlation between model stability (in terms of hyperparameter sensitivity and initialization) and performance suggests that stability metrics could be valuable indicators of model quality. This relationship warrants further investigation and could inform new model selection criteria.

**Limitations**   While our study provides key insights into the role of geometry in Graph Neural Networks, some limitations remain.

First, we observe a strong correlation between certain dataset characteristics, such as heterophily, size, and curvature, making it difficult to isolate the independent effects of these factors on model performance. Future work could address this limitation by investigating dynamic graph datasets, where controlled modifications to specific properties could provide a more granular understanding of their impact on model behavior.

Finally, our study does not account for the computational cost associated with different geometries. Future work should analyze the trade-offs between model performance and computational efficiency to provide more practical guidelines for choosing between Euclidean, hyperbolic, and alternating geometry models.

In conclusion, our comprehensive analysis reveals that the impact of geometry on Graph Neural Networks is highly task-dependent and interacts complexly with architecture choice and dataset characteristics. These findings underscore the importance of careful consideration of these factors in model design and selection, and open up several promising avenues for future research in geometric deep learning.

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

## APPENDIX

## A   Node Classification: Further Results

In this section of the appendix, we include further results from our node classification that were not detailed in our main paper.

### A.1   Best Configurations

In Section 5 of the main paper, we studied the best configurations of each combination of model and dataset, then aggregating results to report accuracy per architecture and geometry in Table 3. Here, we aggregate by geometry and architecture, evaluating results on each dataset.

#### A.1.1   Aggregating Over Architecture

As we noted in the main paper, For the large datasets - Citeseer, Cora, Pubmed, Airports USA, Airports Europe, we see relatively smaller differences between the performances of each geometry. For the smaller datasets however, we see that hyperbolic models specifically struggle, performing on average 10% worse than the other geometries for the WebKB-Texas dataset for instance.

|     | Brazil | Europe | USA | Citeseer | Cora | Pubmed | Cornell | Texas | Wisconsin |
|-----|--------|--------|-----|----------|------|--------|---------|-------|-----------|
| Euc | $\mathbf{36 \pm 2}$ | $42 \pm 1$ | $\mathbf{48 \pm 1}$ | $65 \pm 0$ | $79 \pm 0$ | $75 \pm 0$ | $\mathbf{48 \pm 1}$ | $63 \pm 1$ | $54 \pm 1$ |
| Alt | $34 \pm 2$ | $\mathbf{45 \pm 1}$ | $47 \pm 0$ | $65 \pm 0$ | $79 \pm 0$ | $75 \pm 0$ | $46 \pm 1$ | $\mathbf{65 \pm 1}$ | $\mathbf{58 \pm 1}$ |
| Hyp | $29 \pm 2$ | $44 \pm 1$ | $47 \pm 0$ | $\mathbf{67 \pm 0}$ | $\mathbf{80 \pm 0}$ | $\mathbf{77 \pm 0}$ | $43 \pm 0$ | $54 \pm 1$ | $50 \pm 0$ |

Table 9: Average accuracy and standard error for the different geometry models for each dataset. The average is for the best configuration. Hence we average over all ten trials and three architecture types. The best value in each column is highlighted in bold.

#### A.1.2   Aggregating Over Geometry

In line with our observations in Section 5 of the main paper, we see the GC models outperforming their GCN and GAT counterparts on the Airports and Web Journal Datasets - those datasets where node homophily is low. On the other hand, we see for the Publications, where node homophily is high, the accuracies are nearly identical across architectures.

|     | Brazil | Europe | USA | Citeseer | Cora | Pubmed | Cornell | Texas | Wisconsin |
|-----|--------|--------|-----|----------|------|--------|---------|-------|-----------|
| GCN | $25 \pm 1$ | $44 \pm 1$ | $47 \pm 0$ | $\mathbf{66 \pm 0}$ | $\mathbf{79 \pm 0}$ | $\mathbf{76 \pm 0}$ | $45 \pm 0$ | $\mathbf{61 \pm 1}$ | $51 \pm 0$ |
| GC  | $\mathbf{40 \pm 1}$ | $\mathbf{46 \pm 1}$ | $\mathbf{50 \pm 0}$ | $\mathbf{66 \pm 0}$ | $\mathbf{79 \pm 0}$ | $\mathbf{76 \pm 0}$ | $\mathbf{47 \pm 1}$ | $\mathbf{61 \pm 1}$ | $\mathbf{61 \pm 1}$ |
| GAT | $34 \pm 2$ | $41 \pm 1$ | $46 \pm 0$ | $65 \pm 0$ | $\mathbf{79 \pm 0}$ | $75 \pm 0$ | $45 \pm 0$ | $60 \pm 1$ | $51 \pm 1$ |

Table 10: Average accuracy and standard error for the different architecture models for each dataset. The average is for the best configuration. Hence we average over all ten trials and three geometry types. The best value in each column is highlighted in bold.

## A.2 All Configurations

In Table 2 of the main paper, we aggregated the average accuracy of the best configuration for each model on each dataset. Here, we aggregate the average accuracy across all configurations and report our results in Table 11.

| Dataset | Euclidean | | | Alternating | | | Hyperbolic | | |
|---|---|---|---|---|---|---|---|---|---|
| | GCN | GC | GAT | GCN | GC | GAT | GCN | GC | GAT |
| Airports Brazil | $18 \pm 0$ | $35 \pm 0$ | $30 \pm 1$ | $19 \pm 0$ | $33 \pm 0$ | $36 \pm 0$ | $26 \pm 0$ | $24 \pm 1$ | $22 \pm 0$ |
| Airports Europe | $31 \pm 0$ | $41 \pm 0$ | $37 \pm 0$ | $35 \pm 0$ | $42 \pm 0$ | $39 \pm 0$ | $41 \pm 0$ | $47 \pm 0$ | $38 \pm 0$ |
| Airports USA | $39 \pm 0$ | $46 \pm 0$ | $41 \pm 0$ | $45 \pm 0$ | $45 \pm 0$ | $44 \pm 0$ | $43 \pm 0$ | $45 \pm 0$ | $42 \pm 0$ |
| Citeseer | $58 \pm 0$ | $58 \pm 0$ | $61 \pm 0$ | $60 \pm 0$ | $58 \pm 0$ | $58 \pm 0$ | $60 \pm 1$ | $47 \pm 1$ | $58 \pm 0$ |
| Cora | $70 \pm 1$ | $72 \pm 0$ | $73 \pm 0$ | $75 \pm 0$ | $74 \pm 0$ | $74 \pm 0$ | $70 \pm 1$ | $57 \pm 1$ | $75 \pm 0$ |
| Pubmed | $69 \pm 1$ | $74 \pm 0$ | $71 \pm 1$ | $74 \pm 0$ | $73 \pm 0$ | $73 \pm 0$ | $72 \pm 1$ | $64 \pm 1$ | $72 \pm 0$ |
| Cornell | $45 \pm 0$ | $49 \pm 0$ | $46 \pm 0$ | $45 \pm 0$ | $48 \pm 0$ | $45 \pm 0$ | $44 \pm 0$ | $44 \pm 0$ | $42 \pm 0$ |
| Texas | $60 \pm 0$ | $62 \pm 0$ | $57 \pm 0$ | $61 \pm 0$ | $66 \pm 0$ | $58 \pm 0$ | $52 \pm 0$ | $52 \pm 0$ | $52 \pm 0$ |
| Wisconsin | $52 \pm 0$ | $57 \pm 0$ | $52 \pm 0$ | $54 \pm 0$ | $61 \pm 0$ | $52 \pm 0$ | $46 \pm 0$ | $48 \pm 0$ | $46 \pm 0$ |

Table 11: Average accuracy of all hyperparamter configurations for each model on each dataset along with the standard deviation. Each entry is the average of ten trials. The darker cell color refers to better accuracy.

## A.3 Accuracy Across Hyperparameter Configurations

In this section, we study the accuracies across all hyperparameter configurations for each geometry. We split our experiments by dataset, aggregate over architecture, and plot the accuracy for all hyperparameter configurations as a histogram to study their distribution. The results can be found in Figure 14.

For the Airport datasets, we see that the Euclidean models have a wide range as compared to the spread of hyperbolic and alternating models, which are more concentrated. For Publications, on the other hand, we see that hyperbolic models have the widest spread with several configurations yielding less than 40% accuracy. This is despite the fact that hyperbolic models yielded the highest accuracy for each of the publications datasets at 67%, 79.6%, and 75.6% for Citeseer, Cora, Pubmed respectively. For the Web Journals, we see that hyperbolic models are clustered around lower values. This is reflective of the fact that hyperbolic models have the lowest accuracies for each of these datasets.

## A.4 Sensitivity to Initialization

Here, we tabulate the average standard error for each combination of model and dataset for our node classification experiments. We measure each model's sensitivity to initialization or, in other words, the deviation of accuracy across trials.

Here, we see that alternating models are the most stable across geometries for each architecture. For GCN models, we see that the alternating models have the lowest deviation for 7 out of the 9 datasets, and are significantly less sensitive to initialization that hyperbolic and Euclidean models. For GC and GAT models on the other hand, we see that results are less consistent. For GC, we see that alternating has the lowest standard error in 3 out of the 9 datasets. Hyperbolic models have

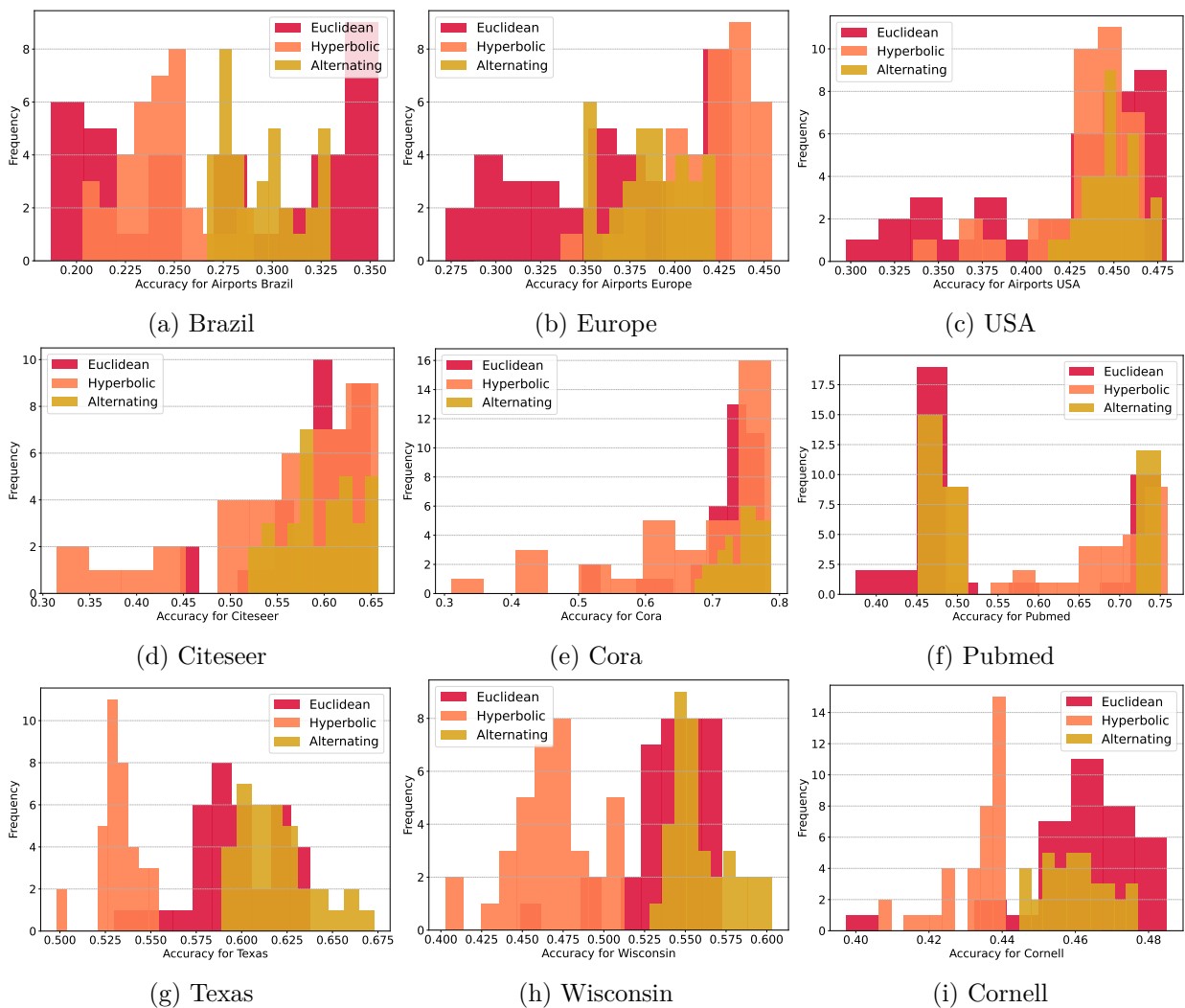

Figure 14: Figure showing the distribution of the mean accuracy (averaged over model arhcitecture) for the 36 different configurations.

the lowest error for all three web journal datasets, but have the highest error for the publications. For the GAT results, we see that alternating models once again have the lowest error in 6 datasets, particularly performing well on the publication datasets.

## A.5 Computational Complexity

During our training for our node classification experiments, we store the convergence epoch - the epoch where validation accuracy is first maximized. We then average this value across all trials of all configurations and report our findings by model and dataset in Table 13

For GCN and GC, we see that alternating models have the smallest convergence epoch on average. With GAT on the other hand, alternating models have the largest convergence epoch while hyperbolic models have the smallest convergence epochs. For all three architectures on the publications, we see that alternating models have the smallest convergence epochs. For the Airports, we see

| Dataset | GCN | | | GC | | | GAT | | |
|---|---|---|---|---|---|---|---|---|---|
| | Euc | Alt | Hyp | Euc | Alt | Hyp | Euc | Alt | Hyp |
| Airports Brazil | 3.09 | **2.18** | 4.08 | 5.97 | **5.09** | 6.14 | 10.47 | 4.56 | **2.96** |
| Airports Europe | 8.01 | 6.54 | **5.03** | 4.96 | 3.91 | **1.96** | 3.51 | **2.31** | 2.79 |
| Airports USA | 6.81 | **1.89** | 4.8 | 4.71 | **2.7** | 2.88 | 6.2 | 2.22 | **2.15** |
| Citeseer | 7.86 | **3.42** | 11.09 | **3.9** | 5.11 | 14.91 | 3.75 | **3.46** | 5.53 |
| Cora | 12.13 | **2.22** | 15.66 | **3.66** | 4.64 | 19.22 | 5.7 | **2.84** | 6.63 |
| Pubmed | 8.91 | **0.59** | 7.34 | 3.77 | **0.87** | 12.04 | 4.41 | **1.1** | 2.53 |
| Cornell | **0.58** | 0.8 | 0.63 | 3.3 | 2.7 | **0.35** | 2.23 | **0.82** | 2.32 |
| Texas | 2.32 | **1.27** | 1.41 | 4.16 | 3.81 | **1.24** | 1.76 | 2.28 | 1.84 |
| Wisconsin | 2.96 | **1.98** | 4.16 | 3.59 | 3.59 | **1.92** | 2.29 | **1.99** | 2.49 |
| Mean | 5.85 | **2.32** | 6.02 | 4.22 | **3.6** | 6.74 | 4.48 | 2.4 | 3.25 |

Table 12: Average standard error for the different architecture models for each dataset. The average is taken over all configurations. Hence, we average over all ten trials and thirty-six configurations. The best value in each column is highlighted in bold.

| Dataset | GCN | | | GC | | | GAT | | |
|---|---|---|---|---|---|---|---|---|---|
| | Euc | Alt | Hyp | Euc | Alt | Hyp | Euc | Alt | Hyp |
| Airports Brazil | 36.6 | 50.6 | **25.2** | 42.0 | **15.5** | 30.7 | 40.1 | 56.0 | **18.3** |
| Airports Europe | 50.7 | 57.6 | **45.5** | 55.1 | **20.7** | 56.6 | 42.4 | 47.9 | **32.5** |
| Airports USA | 61.3 | 60.0 | **52.2** | 65.6 | **19.3** | 53.8 | 57.8 | 58.2 | **45.2** |
| Citeseer | 39.1 | **18.7** | 57.9 | 57.5 | **30.7** | 59.0 | 30.8 | **22.2** | 44.9 |
| Cora | 53.0 | **27.7** | 60.7 | 61.1 | **44.2** | 54.7 | 37.3 | **27.5** | 45.4 |
| Pubmed | 60.9 | **26.3** | 70.5 | 72.5 | **40.2** | 59.9 | 56.7 | **24.9** | 48.3 |
| Cornell | **7.9** | 9.4 | 11.4 | 33.2 | 16.2 | **4.8** | 32.3 | 45.4 | **2.8** |
| Texas | 30.5 | 24.8 | **21.0** | **20.4** | 21.8 | 37.7 | **17.4** | 35.5 | **17.4** |
| Wisconsin | 26.4 | **18.2** | 39.5 | 38.4 | **28.3** | 31.7 | 29.0 | 36.5 | **21.0** |
| Mean | 40.7 | **32.6** | 42.7 | 49.5 | **26.3** | 43.2 | 38.2 | 39.4 | **30.6** |

Table 13: Convergence epochs per experiment. Calculated by averaging the convergence epoch across all trials for all configurations of a combination of dataset and model. For each architecture, the lowest convergence epoch on each dataset is highlighted in bold. The mean row shows the average convergence epoch across all datasets.

that each of the architectures has a different geometry that has the lowest convergence epoch. The results for the web journals on the other hand are varied.

## B   Link Prediction: Further Results

In this section, we include further results from our link prediction experiments, similar to the node classification results explored in Appendix A that were not included in the link prediction analysis in Section 6.

## B.1 Best Configurations

Similar to our reports in Appendix A.1, we aggregate our results of best configurations over geometry and architecture, evaluating results on each dataset

### B.1.1 Aggregating Over Architecture

Here, we see that hyperbolic models have the strongest performance on 7 out of our 9 datasets, in line with our assessment in the main paper. We see that the for smaller datasets, the web journals in particular, that Euclidean models have an advantage on average over hyperbolic models. For the larger datasets, hyperbolic continues to have an advantage.

|     | Brazil | Europe | USA | Citeseer | Cora | Pubmed | Cornell | Texas | Wisconsin |
|-----|--------|--------|-----|----------|------|--------|---------|-------|-----------|
| Euc | $85 \pm 0$ | $84 \pm 0$ | $90 \pm 0$ | $93 \pm 1$ | $95 \pm 0$ | $92 \pm 0$ | $\mathbf{77 \pm 1}$ | $\mathbf{76 \pm 1}$ | $80 \pm 1$ |
| Alt | $75 \pm 2$ | $76 \pm 2$ | $79 \pm 2$ | $76 \pm 1$ | $74 \pm 2$ | $81 \pm 1$ | $67 \pm 2$ | $64 \pm 2$ | $61 \pm 2$ |
| Hyp | $\mathbf{89 \pm 1}$ | $\mathbf{87 \pm 1}$ | $\mathbf{95 \pm 0}$ | $\mathbf{99 \pm 0}$ | $\mathbf{99 \pm 0}$ | $\mathbf{98 \pm 0}$ | $\mathbf{77 \pm 2}$ | $71 \pm 2$ | $\mathbf{82 \pm 1}$ |

Table 14: Average AUC and standard error for the different geometry models for each dataset for link prediction. The average is for the best configuration. Hence, we average over all ten trials and three architecture types. The best value in each column is highlighted in bold.

### B.1.2 Aggregating Over Geometry

Here, we take a closer look at the advantage that GC models have over the architectures for our Link Prediction experiments, outperforming the other architectures in nearly every dataset. We also observe that the margin of this advantage is consistent across all datasets.

|     | Brazil | Europe | USA | Citeseer | Cora | Pubmed | Cornell | Texas | Wisconsin |
|-----|--------|--------|-----|----------|------|--------|---------|-------|-----------|
| GCN | $82 \pm 2$ | $81 \pm 2$ | $87 \pm 2$ | $89 \pm 2$ | $89 \pm 2$ | $90 \pm 2$ | $69 \pm 2$ | $70 \pm 2$ | $72 \pm 3$ |
| GC  | $\mathbf{84 \pm 0}$ | $\mathbf{84 \pm 0}$ | $\mathbf{92 \pm 0}$ | $\mathbf{91 \pm 2}$ | $\mathbf{90 \pm 2}$ | $\mathbf{92 \pm 1}$ | $\mathbf{76 \pm 2}$ | $70 \pm 2$ | $\mathbf{78 \pm 1}$ |
| GAT | $82 \pm 2$ | $82 \pm 1$ | $85 \pm 2$ | $88 \pm 2$ | $88 \pm 3$ | $88 \pm 2$ | $76 \pm 3$ | $\mathbf{71 \pm 2}$ | $72 \pm 2$ |

Table 15: Average AUC and standard error for the different architecture models for link prediction. The average is for the best configuration. Hence, we average over all ten trials and three geometry types. The best value in each column is highlighted in bold.

## B.2 AUC for all configurations

In Table 5, we explored the accuracy of the best configuration for each model on each dataset. In this section, we will tabulate a similar table, averaging over all hyperparameter configurations.

We first note that alternating GCN and GAT models have significantly lower AUC scores than what we saw with best hyperparameter configuration in Table 5. Alternating GC on other hand, does not see the same levels of reduction in performance. The hyperbolic models, on the Publications and Airports USA in particular, have very minimal differences in AUC as compared to best configuration. Euclidean models have the greatest differences, with Euclidean GC on Cora and Citeseer seeing drops in 13% and 14% respectively. The Euclidean GC models are the most sensitive to hyperparameter out of the Euclidean models.

| Dataset | Euclidean | | | Alternating | | | Hyperbolic | | |
|---|---|---|---|---|---|---|---|---|---|
| | GCN | GC | GAT | GCN | GC | GAT | GCN | GC | GAT |
| Airports Brazil | $76 \pm 0$ | $84 \pm 0$ | $79 \pm 0$ | $57 \pm 0$ | $82 \pm 0$ | $58 \pm 0$ | $89 \pm 0$ | $83 \pm 0$ | $91 \pm 0$ |
| Airports Europe | $78 \pm 0$ | $80 \pm 0$ | $80 \pm 0$ | $57 \pm 0$ | $82 \pm 0$ | $65 \pm 1$ | $83 \pm 0$ | $81 \pm 0$ | $84 \pm 0$ |
| Airports USA | $86 \pm 0$ | $85 \pm 0$ | $83 \pm 0$ | $60 \pm 0$ | $89 \pm 0$ | $61 \pm 0$ | $94 \pm 0$ | $91 \pm 0$ | $95 \pm 0$ |
| Citeseer | $89 \pm 0$ | $82 \pm 0$ | $86 \pm 0$ | $60 \pm 1$ | $74 \pm 0$ | $60 \pm 0$ | $98 \pm 0$ | $98 \pm 0$ | $98 \pm 0$ |
| Cora | $88 \pm 0$ | $81 \pm 1$ | $90 \pm 0$ | $58 \pm 0$ | $73 \pm 0$ | $57 \pm 0$ | $98 \pm 0$ | $97 \pm 0$ | $98 \pm 0$ |
| Pubmed | $83 \pm 1$ | $91 \pm 0$ | $83 \pm 1$ | $62 \pm 1$ | $86 \pm 0$ | $60 \pm 1$ | $97 \pm 0$ | $95 \pm 0$ | $99 \pm 0$ |
| Cornell | $69 \pm 0$ | $80 \pm 0$ | $74 \pm 1$ | $57 \pm 0$ | $80 \pm 0$ | $55 \pm 0$ | $77 \pm 0$ | $58 \pm 0$ | $86 \pm 0$ |
| Texas | $68 \pm 0$ | $75 \pm 0$ | $72 \pm 0$ | $54 \pm 0$ | $74 \pm 0$ | $54 \pm 0$ | $75 \pm 0$ | $58 \pm 0$ | $78 \pm 0$ |
| Wisconsin | $75 \pm 0$ | $75 \pm 0$ | $71 \pm 0$ | $51 \pm 0$ | $69 \pm 0$ | $52 \pm 0$ | $80 \pm 0$ | $76 \pm 0$ | $87 \pm 0$ |

Table 16: Average AUC of each model averaged over all hyperparameters along with the standard deviation. The darker cell color refers to better AUC.

### B.3 Accuracy Across Hyperparameter Configurations

As we did in Appendix A.3, for our link prediction experiments, we plot the distribution of AUC scores by hyperparameter configuration for each geometry.

In comparison to the node classification distributions in Table 14, we see a narrower spread for all three geometries. The hyperbolic models in particular have the smallest range. For instance, for Cora and Citeseer, we see all hyperbolic models performing within a 5% range. The alternating models appear to have the widest spread while also having the lowest AUCs. We note that the Publications have the most varied AUCs, with the hyperbolic once again having the most concentrated results.

### B.4 Sensitivity to Initialization

| Dataset | GCN | | | GC | | | GAT | | |
|---|---|---|---|---|---|---|---|---|---|
| | Euc | Mix | Hyp | Euc | Mix | Hyp | Euc | Mix | Hyp |
| Airports Brazil | 5.34 | 5.27 | **4.27** | 1.65 | **0.95** | 1.3 | 5.12 | 6.69 | **0.55** |
| Airports Europe | **4.84** | 5.08 | 8.21 | 6.44 | 3.11 | **1.43** | **0.94** | 8.48 | 1.94 |
| Airports USA | 3.44 | 6.62 | **2.34** | 5.06 | 1.34 | **1.07** | 3.25 | 6.59 | **0.55** |
| Citeseer | 2.71 | 8.16 | **0.84** | 8.64 | 1.84 | **0.66** | 2.56 | 8.27 | **0.51** |
| Cora | 5.61 | 6.89 | **1.28** | 10.09 | 2.3 | **1.17** | 2.57 | 6.33 | **0.32** |
| Pubmed | 7.52 | 11.1 | **1.73** | 3.26 | **0.69** | 0.94 | 5.2 | 10.14 | **0.54** |
| Cornell | 5.42 | 5.31 | **4.05** | 2.78 | 4.26 | **2.6** | 6.82 | 3.38 | **0.87** |
| Texas | 5.91 | 4.07 | **2.7** | 1.59 | 2.32 | **1.54** | 3.24 | 4.1 | **1.52** |
| Wisconsin | 7.22 | 2.75 | **2.44** | 6.06 | 2.67 | **1.58** | 7.91 | 2.35 | **0.91** |
| Mean | 5.33 | 6.14 | **3.09** | 5.06 | 2.16 | **1.36** | 4.18 | 6.26 | **0.86** |

Table 17: Average standard error for the different architecture models for each dataset. The average is taken over all configurations, i.e., overall ten trials and thirty-six configurations. The best value in each column is highlighted in bold.

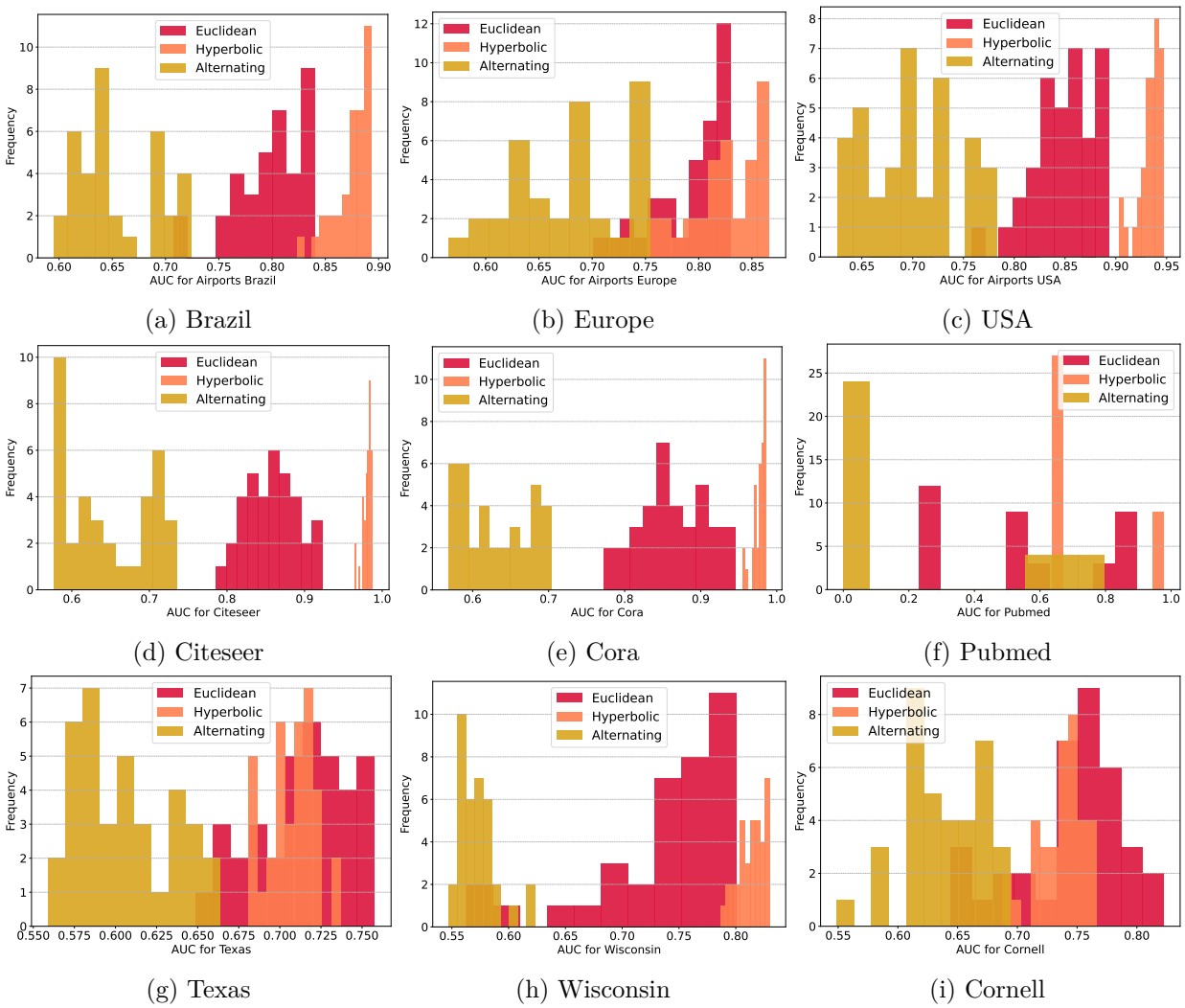

Figure 15: Figure showing the distribution of the mean accuracy (averaged over model architecture) for the 36 different configurations.

Here, we repeat the experiments that were outlined in Appendix A.4, where we tabulated the average standard error for each combination of model and dataset. Unlike in our node classification experiment, we observe notable trends in the link prediction task, as shown in Table 17.

From the table, it is clear that hyperbolic models consistently exhibit the lowest standard error across architectures and datasets, indicating their superior stability. Specifically, the hyperbolic GAT stands out as the most stable model overall, achieving the lowest standard error in nearly all datasets. Notably, it also achieves the highest mean AUC across all nine models, further reinforcing its robustness and reliability.

Conversely, we observe that alternating GCN and GAT models display the highest variability in standard error across trials. These two models also report the lowest mean AUC, suggesting that their instability may correlate with their comparatively weaker performance. The Euclidean models show moderate stability, with standard errors typically higher than their hyperbolic counterparts

but lower than alternating models, with the exception of the GC architecture, where the Euclidean models are the least stable.

## B.5  Computational Complexity

Similar to our reports in Section A.5 of the Appendix, we collect the convergence epoch for each of our experiments, average our results for all trials of all configurations and report our findings for each model on each datasets.

| Dataset | GCN | | | GC | | | GAT | | |
|---|---|---|---|---|---|---|---|---|---|
| | Euc | Alt | Hyp | Euc | Alt | Hyp | Euc | Alt | Hyp |
| Airports Brazil | 141.5 | **17.2** | 162.2 | 102.9 | 49.3 | **24.2** | 136.7 | **30.3** | 162.1 |
| Airports Europe | 138.5 | **14.2** | 195.5 | 127.1 | 128.4 | **3.0** | 145.0 | **18.7** | 187.1 |
| Airports USA | 189.0 | **16.3** | 194.7 | 92.0 | 132.6 | **18.5** | 161.1 | **15.8** | 188.3 |
| Citeseer | 145.5 | **26.4** | 136.2 | 129.9 | 110.4 | **19.6** | 173.4 | **33.0** | 135.7 |
| Cora | 185.0 | **30.5** | 161.1 | 131.8 | 146.2 | **16.5** | 176.4 | **31.7** | 165.6 |
| Pubmed | 195.2 | **61.3** | 174.2 | 170.7 | **14.5** | 38.8 | 190.5 | **58.9** | 194.2 |
| Cornell | 39.2 | **22.8** | 69.9 | 74.1 | 97.8 | **44.0** | 83.6 | **14.9** | 62.4 |
| Texas | 64.4 | **13.3** | 68.3 | **26.8** | 68.3 | 45.2 | 66.5 | **28.2** | 79.3 |
| Wisconsin | 108.8 | **18.1** | 91.1 | **79.4** | 100.9 | 81.5 | 129.6 | **29.6** | 87.5 |
| Mean | 134.1 | **24.5** | 139.2 | 103.9 | 94.3 | **32.4** | 140.3 | **29.0** | 140.2 |

Table 18: Convergence epochs per experiment. Calculated by averaging the convergence epoch across all trials for all configurations of a combination of dataset and model.

We observe that for the GAT and GCN models, alternating models train the fastest, with the Eucldiean and hyperbolic converging at comparable rates on average. Recall that for these architectures, alternating models produce the lowest AUCs, whereas hyperbolic the highest. For GC models on the other hand, we see that hyperbolic models have the smallest convergence epoch for 6 out of the 9 datasets, converging significantly faster than the hyperbolic GAT and GCN models. It is also worth noting that the hyperbolic GC models yield the lowest AUC scores out of all hyperbolic models.

