# OpenReview forum: "Effect of Geometry on Graph Neural Networks"
_TMLR — Rejected by TMLR_

### Review · Reviewer_3phN · 2025-07-22

**Summary Of Contributions:**

Graph neural networks are a popular modeling paradigm for graph structured data, and have enjoyed popularity in both research and industrial settings for exactly this purpose. It is common to view a graph as a discrete topology, and working from this view has revealed many interesting directions for graph based machine learning.  The literature has focused on many different techniques to explicitly model graph geometry, including hyperbolic gnns and Ricci curvature aware gnns, which have yielded at times impressive results. The authors of this work continue in this direction and develop a geometrically aware message passing gnn. To develop their method the authors explore a variety of different approaches to capturing graph geometry in message passing and explore the relationship between geometric hyper parameters and graph structure. Finally, the authors validate these intuitions, and their models, on two different standard graph tasks -- Node classification and Link prediction.

**Audience:**

Yes

**Claims And Evidence:**

No

**Requested Changes:**

Please see the weaknesses section

**Strengths And Weaknesses:**

**Strengths**
1. The paper is clearly written
2. The experiments are well presented
3. The analysis and meta-analysis is exhaustive
4. The alternating geometry approach avoids the need for Riemannian optimization, which could make it more accessible to practitioners.
5. The code is publicly available

**Weaknesses**
1. The euclidean results reported in table 2 don't appear to comport with published values. For example, a GCN on Cora should achieve a node classification accuracy of ~88%. All euclidean baselines indeed appear to be much lower than published values, calling into question the results published here.
2. The presented datasets are all quite small, and of limited scientific value. For example, the webkb datasets are known to be flawed due to their small size, and strange statistics (https://arxiv.org/pdf/2302.11640).
3. The motivation of the alternating geometry model is unclear. Why should this particular mapping (Equation 2) be optimal or even beneficial? What explains the divergence from Chen et al?
4. The poincare disk model seems to be ignored, despite its popularity. Why?
5. The computational costs of the method don't appear to be discussed. Neither do scalability issues.
6. Numerical stability/stabilization concerns are not discussed.
7. Most claims of out performance appear to be gains within error bars. The authors might benefit from providing a rank test.
8. Many missing baselines that might be relevant for both link prediction and node classification. Examples include H2GCN, GraphSAGE, GIN, HGCN, ELPH, BUDDY, SEAL, etc. Other relevant literature could include Hypformer and equivalent work.
9. Some results are difficult to interpret (e.g., very low link prediction AUCs in Table 5) and dismissed as "outliers" without adequate explanation. Please expand
10. No explanation of convergence properties of different geometries.
11. The chosen datasets make it difficult to isolate individual effects, which makes it difficult to know how much of sections 5 and 6 will generalize.
12. Different geometries may have fundamentally different optimal hyperparameter ranges, making direct comparison problematic. The same grid search across all methods may systematically disadvantage some approaches.
13. Limited discussion of train/val/test splits or potential data leakage
14. No systematic ablation studies
15. Limited discussion of when geometric considerations actually matter vs. when simpler Euclidean methods suffice.
16. No discussion of other geometries that would be potentially relevant (eg spheres) or even in product spaces.

---

### Review · Reviewer_Egi4 · 2025-09-01

**Summary Of Contributions:**

This paper introduces the idea of alternating geometry in the context of hyperbolic feature geometry.
It discusses extensive experiments about the impact of choice of feature geometry (Euclidean, hyperbolic, and the proposed one, alternating) and draws some qualitative conclusions about network design as a function of task (dataset features x task type).
The new way to handle geometry that is introduced is not conclusively superior to pre-existing ones, yet the experiments are informative about how the choice of how to handle feature geometry impacts results, depending on the task at hand.
The discussions of results are sometimes a bit lengthy, but summaries for each part and an overall summary are concise and informative.
It is written clearly, with a clear and sensible structure.

**Audience:**

Yes

**Broader Impact Concerns:**

This work is leaning on the Red AI side, with extensive experiments and limited intuitive understanding of the output results.
However, the realease of all the fine-grained experimental data, which may be further leveraged by manual analysis and/or by automated methods such as in auto-DL, is a very positive point.

**Claims And Evidence:**

Yes

**Requested Changes:**

# Important changes
- The intuition over how much the datasets used are hyperbolic (OCR and Gromov delta) should be mentionned early. In the current version, it is a bit mystifying to the reader at first, how citation networks may be hyperbolic or not, positively or negatively curved.
- Also, authors should try to give a few intuitions over what positive or negative ORC and positive delta mean, for the data. Is hyperbolicity related to the fact that networks tend to have hubs (nodes with very high degrees, or at least clusters of nodes that connect to many others?). I am new to this topic and it is not immediately clear, a few sentences of qualitative explanation would help. Also, explain how both measures capture different things, what is the nunance between them.
- Optional, but could be nice for self-sufficiency of the paper: in the appendix, state the definitions of ORC and Gromov delta, with equations.
- "There could be difference in the expressivity for the different geometries." Please add a couple of lines to explain how the hyperbolic vs Euclidean geometries impact the effective expressivity of an architecture. Is the parameter count changing, or not ? Or is it a more subtle change, in the sense that the function space is distorted (maybe restricted in the hyperbolic case, consistent with the ideas in general geometric deep learning, where accounting for symmetries helps the network learn, despite strictly reducing the hypothesis space?)
- Table 2 (and same remark for table 5): I understand that these raw results are then discussed in detail. However, I think the current colormap is very uninformative. I would either suppress it, or, better, replace it with a row-wise colormap: for each row, normalize the colormap between best and worse performance, and color proportionally to the performance *for this dataset*.
Or, adopt the same coding as for table 3,4,6,8: best column in each row being shown in bold (ties can be all in bold).
Currently, it just tells us how the models studied do on each dataset (task), i.e. some proxy to the intrinsic difficulty of each task, which is irrelevant.
- "However, with the overlapping
uncertainty intervals for the accuracy, we cannot definitively claim that the hyperbolic GC outper-
forms those models." ->
- Section 5.4.0 (before 5.4.1) (and 6.5.0, i.e. the part of 6.5 before 6.5.1): move to appendix, together with figures 6,7,12,13 ? Indeed I don't really see the take-home message from these, apart that small depth and large width are often preferred (probably saturating the range of values that were allowed, as mentionned above). Keep the sensitivity analysis, which is indeed helpful.
- 5.4.1 and 6.5.1, Sensitivity to Hyperparameter: nice sections. Could be improved by displaying, in table 4b and its counterpart in sec 6, the relative variation between average in all and average in best configs. (e.g. best/all, i.e. a number larger than 1, where larger values correspond to more sensitive models, and a value of 1, to a very robust model). You may as well get rid of the tables (a), and possibly add the standard deviation between correlation values of the "all configs" setup, where again a large value would indicate sensitivity.
Same for table 11.
- section "Limitations": Indeed, if you could report the average GPU hours used along with the epochs, in the appendix, that could be nice (I guess that for a fixed number of epochs, each geometry and architecture has a given compute budget - actually one per task).
- section "Limitations": I would also make a link with Auto-DL. The fact you kindly release the fine-grained experimental results may pave the way for auto-DL methods that automatically chooses hyperbolic geometry and architecture type.


# Small changes and Typos


- section 3, simplified version of Chen 2022. Could you explicit how your choice of approximation of the formula for z (last equation of section 2) relates to the case $v \to 0$ ? Intuitively it seems that there may be a link.
- Olivier Ricci Curvature -> Ollivier Ricci Curvature
- page 10: δ-hyperbolicity . Page 13: Gromov hyperbolicity. Please make the naming consistent across the paper. I stumbled on Gromov on page 13 because I felt like it was a new notion.
- "We note that our consists of two parts the graph and the feature vectors" -> a word or a couple of words is missing.
- "This provides two different loss geometries for comparison." this sentence seems empty of information to me, you may delete it.
- Over-use and most importantly, some mis-uses of "Hence". Please rephrase, esp. in the sections discussing the results. Simpler words like "Also", "Furthermore", or simply starting a new sentence, often makes more sense.
- figure 2: the negative comes first, which makes sense. In the text, the positive is discussed first. Please swap the discussion of positive and negative in the text.
- "The belief used to be" -> [citation needed]
- Typo: "However, on low datasets with low (...)".  ??
- table 7: why is hyperbolic the first and not the last column, as usual ? It's a bit unsettling, please change it.
- section 6.3: this is an occurrence of "not so much to say" where the text feels needlessly long, because there is no strking result to report. I would consider making it more concise (optional)
- "This paves the way for further research into the theoretical
foundations of interactions between geometry and architecture" -> incorrect, because the paper does not start any theoretical analysis: it is a very empirical approach (which may help trigger theoretical works, yes). You open up a theoretical question.

**Strengths And Weaknesses:**

# Strengths:
- Clear introductory parts
- publication of the fine-grained experimental data
- very clear, legible paper structure, summary of take-home messages in each part and at the paper level.
- rather extensive experiments (multiple architectures, 10 trials each time, etc)

# Weaknesses
- sometimes irrelevant discussions, discussing exhaustively the numerical results, even when there is no striking fact (a bit lengthy to read)
- Important problem to deal with: Comparing tables 5 and 18, I suspect that there may be an overfitting (early stopping occurring early) for the models that perform badly. A simple scatter plot of performance vs best epoch could help visualize whether this is true (maybe one color per dataset-task). If performance strongly correlates with best epoch, it may be an indication that poorly-performing models actually suffer from early overfitting, which also induces overall underfitting. A decreased learning rate or some regularization tools may help train better these models (making a better use of the available epochs, probably). If it is the case, some of the qualitative conclusions of the paper may have to be updated.
- numerical experiments: limited max. width, and smallest depth, although the best values are often the smallest depth (2) and max width (256), indicating a possible saturation of the range of values explored. Possibly same remark for learning rates.
- these 2 points make the "Claims And Evidence" Yes/No choice lean on No.
- numerical experiments: averaging over architectures seems sometimes irrelevant. Architecture choice can be seen as a (meta, high-level) hyperparameter to tune, and the best value should be chosen. This is especially true when performances for a given feature geometry choice and various architectures differ widely, as in the link prediction task. One could compare the trends of the various geometries by comparing the best performing architecture of each, rather than the average over 3 architecture (which are 3 potentially qualitatively different things).

---

### Review · Reviewer_WkHP · 2025-10-15

**Summary Of Contributions:**

The paper provides two main contributions:

1. **A new GNN architecture (alternating-GNNs)** that combines Euclidean-GNNs and Hyperbolic-GNNs into a unified model. This approach eliminates the need for specialized aggregation in hyperbolic space by introducing appropriate linear transformations, allowing the use of standard message-passing aggregation and gradient-based optimization instead of Riemannian gradients.

2. **A detailed experimental analysis** exploring how the following factors affect performance:
   - The geometry of the graph
   - The GNN architecture
   - The model geometries (Euclidean, hyperbolic, or alternating). The paper calls this last factor feature geometry.

**Audience:**

Yes

**Broader Impact Concerns:**

I do not identify any immediate broader impact or ethical concerns associated with this work.

**Claims And Evidence:**

Yes

**Requested Changes:**

**Critical**
1. Clarify the assumptions and reasoning under which the Ollivier–Ricci curvature (ORC) is claimed to characterize the geometry of a graph.

2. Improve the clarity and consistency of the various notions of geometry used throughout the paper. Only data geometry and feature geometry seem central to the main comparison and should be emphasized.

3. Explicitly write the equations of the hyperbolic GNN implementations.

4. Adopt more expressive GNN architectures for link-prediction experiments to strengthen the discrimination between the two tasks.


**Optional** (strongly recommended)
1. Use synthetic datasets to better control and manipulate the geometry of the graph via the underlying manifold. This would make it possible to isolate geometric effects, validate the interpretation of ORC, and compare with additional geometric metrics.

2. Using the above synthetic datasets, define proper target functions allowing an evaluation of the different models based on the different target functions. This would provide a more structured comparison between models and tasks, going beyond the general distinction between node-level and edge-level prediction.

**Strengths And Weaknesses:**

**Strengths**

1. The exploration of how the geometry of a graph can inform which type of GNN to use is a highly interesting and important problem that could have practical implications for model selection in geometric deep learning.

2. The experiments are clearly described, and the comparisons appear fair. The authors are also transparent about which numerical differences are significant and how these relate to broader geometric insights. Overall, the experimental results support reasonable and meaningful conclusions.


**Weaknesses**

1. The paper uses the term “geometry” in a broad and sometimes inconsistent way, which makes it difficult to understand the intended meaning until the experimental section. In particular:
   - *Data geometry* (Section 1.1) is defined as the geometry of the manifold on which the graph is supported, but in experiments it is represented using Ollivier–Ricci curvature (ORC). The relationship between these two notions should be made explicit.
   - *Feature geometry* refers to the type of space used for aggregation (Euclidean, hyperbolic, alternating), which differs conceptually from data geometry.
   - *Loss geometry* is used to distinguish node and link prediction tasks via their loss functions, but this definition of geometry is unclear — especially since the two loss functions are mentioned as different but never explicitly presented. I assume cross-entropy loss is used in both tasks.

2. Parameter geometry, mentioned in Section 4.2, is not analyzed and could either be omitted or renamed to avoid confusion among the many geometric notions introduced.

3. For the link prediction tasks, the authors use the inner product of node features to represent edge features. This approach can miss key structural information. More expressive models such as those using the *Labeling Trick* [*Labeling Trick: A Theory of Using Graph Neural Networks for Multi-Node Representation Learning*, M. Zhang et al.] would yield a clearer distinction between node and link prediction models and more meaningful comparisons. As now link prediction is more or less treated as a node prediction task.

4. The geometry of the graph is assessed using Ollivier–Ricci curvature (ORC). However, to make the analysis more convincing and interpretable, it would be beneficial to use synthetic graphs generated from known manifolds. In such cases, ORC would be a more representative metric for the geometry of the graph given its relation with the curvature of the underlying manifold, and additional geometric metrics could be compared to provide stronger evidence.

5. The Hyperbolic GNNs used for comparison are not clearly defined through their explicit equations (unlike the alternating and Euclidean GNNs in Section 3.2).

---

### Decision · Action_Editor_gnZb · 2025-11-14

**Recommendation:** Reject

**Audience:**

Yes

**Audience Explanation:**

The authors have not responded to the raised weaknesses and the detailed requested changes provided by the reviewers. All reviewers agree to reject the paper in its current form; I follow this recommendation.

**Claims And Evidence:**

No

**Claims Explanation:**

See comment below.